# Learning predictive checklists
# with probabilistic logic programming

## Abstract

Checklists have been widely recognized as effective tools for completing complex tasks in a systematic manner. Although originally intended for use in procedural tasks, their interpretability and ease of use have led to their adoption for predictive tasks as well, including in clinical settings. However, designing checklists can be challenging, often requiring expert knowledge and manual rule design based on available data. Recent work has attempted to address this issue by using machine learning to automatically generate predictive checklists from data, although these approaches have been limited to Boolean data. We propose a novel method for learning predictive checklists from diverse data modalities, such as images and time series. Our approach relies on probabilistic logic programming, a learning paradigm that enables matching the discrete nature of checklist with continuous-valued data. We propose a regularization technique to tradeoff between the information captured in discrete concepts of continuous data and permit a tunable level of interpretability for the learned checklist concepts. We demonstrate that our method outperforms various explainable machine learning techniques on prediction tasks involving image sequences, time series, and clinical notes.

## 1 Introduction

In recent years, machine learning models have gained popularity in the healthcare domain due to their impressive performance in various medical tasks, including diagnosis from medical images and early prediction of sepsis from clinical time series, among others (Davenport & Kalakota, 2019; Esteva et al., 2019). Despite the proliferation of these models in the literature, their wide adoption in real-world clinical practice remains challenging (Futoma et al., 2020; Ahmad et al., 2018; Ghassemi et al., 2020; De Brouwer et al., 2022). Ensuring the level of robustness required for healthcare applications is difficult for deep learning models due to their inherent black box nature. Non-interpretable models make stress testing arduous and thus undermine the confidence required to deploy them in critical applications such as clinical practice. To address this issue, recent works have focused on developing novel architectures that are both human-interpretable and retain the high performance of black box models (Ahmad et al., 2018).

One such approach is learning medical checklists from available medical records. A checklist is a discrete linear classifier consisting of binary conditions that predict an outcome. For instance, it can be used to diagnose a disease by determining if at least T out of M symptoms are present in a patient. Due to their simplicity and ability to assist clinicians in complex situations, checklists have become increasingly popular in medical practice (Haynes et al., 2009). However, the simplicity of using checklists typically contrasts with the complexity of their design process. Creating a performant checklist requires domain experts who manually collect evidence about the particular clinical problem of interest, and subsequently reach consensus on meaningful checklist rules (Hales et al., 2008). As the number of available medical records grows, the manual collection of evidence becomes more tedious, bringing the need for partially automated design of medical checklists.

Recent works have taken a step in that direction by learning predictive checklists from Boolean, categorical, or continuous tabular data (Zhang et al., 2021; Makhija et al., 2022). Nevertheless, many available clinical data, such as images or time series, are neither categorical nor tabular by nature. They therefore fall outside

the limits of applicability of previous approaches for learning checklists from data. This work aims at building checklists based on the presence or absence of concepts learnt from high-dimensional data, thereby addressing this limitation.

Prior work leverages integer programming to generate checklists, but the discrete (combinatorial) nature of solving integer programs makes it challenging to learn predictive checklists from images or time series data. Deep learning architectures rely on gradient-based optimization which differs in style and is difficult to reconcile with integer programming (Shvo et al., 2021). We instead propose to formulate predictive checklists within the framework of probabilistic logic programming. This enables extracting binary concepts from high-dimensional modalities like images, time series, and text data according to a probabilistic checklist objective, while propagating derivatives throughout the entire neural network architecture. As a proof-of-concept, we show that our architecture can be leveraged to learn a checklist of decision trees operating on concepts learnt from the input data, highlighting the capacity of our approach to handle discrete learning structures.

Our architecture, ProbChecklist, operates by learning binary concepts from high-dimensional inputs, which are then used for evaluating the checklist. This is unlike existing approaches that rely on summary statistics of the input data (e.g. mean or standard deviation of time series). Nevertheless, this flexibility also reports parts of the burden of interpretability to the learnt concepts. Indeed, depending on how the concepts are learnt, their meaning can become unclear. We therefore investigate two strategies for providing predictive and interpretable concepts. The first relies on using inherently interpretable concept extractors, which only focus on specific aspects of the input data (Johnson et al., 2022). The second adds regularization penalties to enforce interpretability in the neural network by design. Several regularization terms have been coined to ensure the concepts are unique, generalizable, and correspond to distinctive input features (Jeffares et al., 2023) (Zhang et al., 2018).

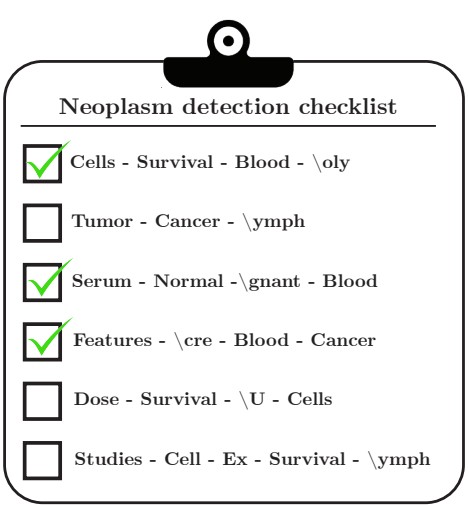

Figure 1: Example checklist learnt by our architecture. Three or more checks entail a positive neoplasm prediction. We identify key tokens in clinical notes that correspond to positive and negative concepts, where each concept is characterized by the presence of positive and absence of negative tokens.

Clinical practice is a highly stressful environment where complex decisions with far-reaching consequences have to be made quickly. In this context, the simplicity, robustness, and effectiveness of checklists can make a difference (Hales et al., 2007). Furthermore, healthcare datasets contain sensitive patient information, including ethnicity and gender, which should not cause substantial differences in the treatment provided. However, machine learning models trained on clinical data have been shown to exhibit unacceptable imbalance of performance for different population groups, resulting in biased predictions. When allocating scarce medical resources, fairness should be emphasized more than accuracy to avoid targeting minority subgroups (Fawzy et al., 2022). In an attempt to mitigate this problem, we study the impact of including a fairness regularization into our architecture and report significant reductions in the performance gap across sensitive populations. We validate our approach empirically on several classification tasks using various data modalities such as images and clinical time series. We show that ProbChecklist outperforms previous learnable predictive checklist approaches as well as interpretable machine learning baselines. We showcase the capabilities of our method on two healthcare case studies: early prediction of sepsis and mortality in the intensive care unit.

**Contributions.**

- We propose the first framework to learn predictive checklists from arbitrary input data modalities. Our approach can learn checklists and extract meaningful concepts from time series and images, among others. In contrast with previous works that used (mixed-)integer programming, our approach

formulates the predictive checklist learning within the framework of probabilistic logical programming. This makes ProbChecklist more amenable to modern deep learning optimization methods.

- We investigate the impact of different schemes for improving the interpretability of the concepts learnt as the basis of the checklist. We employ regularization techniques to encourage the concepts to be distinct, so they can span the entire input vector and be specialized, i.e. ignore the noise in the signal and learn sparse representations. We also investigate the impact of incorporating fairness constraints into our architecture.

- We validate our framework on different data modalities such as images, text and time series, displaying significantly improved performance compared to state-of-the-art checklist learning methods.

## 2 Related works

A major motivation for our work is the ability to learn effective yet interpretable predictive models from data, as exemplified by the interpretable machine learning literature. Conceptually, our method builds upon the recent body of work on learning predictive checklists from data. The implementation of our solution is directly inspired by the literature on probabilistic logic programming.

**Interpretable machine learning.**  Motivated by the lack of robustness and trust of black box models, a significant effort has been dedicated to developing more human-interpretable machine learning models in the last years (Ahmad et al., 2018; Murdoch et al., 2019). Among them, one distinguishes between *intrinsic* (*i.e.* when the model is itself interpretable such as decision trees) and *posthoc* (*i.e.* when trained models are interpreted a posteriori) methods (Du et al., 2019). Checklists belong to the former category as they are an intuitive and easy to use decision support tool. Compared to decision trees, checklists are more concise (there is no branching structure) and can thus be potentially more effective in high stress environments (a more detailed argument is presented in Appendix D). Our approach also relies on building concepts from the input data. Because the concepts are learnt from data, they may themselves lack a clear interpretation. Both intrinsic and posthoc interpretability techniques can then be applied for the concept extraction pipeline (Jeffares et al., 2023). Concept Bottleneck Models (Koh et al., 2020) insert a concept layer before the last fully connected layer, assigning a human-understandable concepts to each neuron. However, a major limitation is that it requires expensive annotated data for predefined concepts.

**Rule-based learning.**  Boolean rule mining and decision rule set learning is a well-studied area that has garnered considerable attention spurred by the demand for interpretable models. Some examples of logic-based models include Disjunctive Normal Forms (OR of ANDs), Conjunctive Normal Forms (AND of ORs), chaining of rules in the form of IF-THEN-ELSE conditions in decision lists, and decision tables. Most approaches perform pre-mining of candidate rules and sample rules using integer programs (IP), simulated annealing, performing local search algorithm for optimizing simplicity and accuracy (Lakkaraju et al., 2016), and Bayesian framework for constructing a maximum a posteriori (MAP) solution (Wang et al., 2017).

**Checklist learning.**  Checklists, pivotal in clinical decision-making, are typically manually designed by expert clinicians (Haynes et al., 2009). Increasing medical records make manual evidence collection tedious, prompting the need for automated medical checklist design. Recent works have taken a step in that direction by learning predictive checklists from Boolean or categorical medical data (Zhang et al., 2021). Makhija et al. (2022) have extended this approach by allowing for continuous tabular data using mixed integer programming. Our work builds upon these recent advances but allows for complex input data modalities. What is more, in contrast to previous works, our method does not rely on integer programming and thus exhibits much faster computing times and is more amenable to the most recent deep learning stochastic optimization schemes.

**Probabilistic logical programming.**  Probabilistic logic reasoning combines logic and probability theory. It represents a refreshing framework from deep learning in the path towards artificial intelligence, focusing on high-level reasoning. Examples of areas relying on these premises include statistical artificial intelligence (Raedt et al., 2016; Koller et al., 2007) and probabilistic logic programming (De Raedt & Kimmig, 2015).

More recently, researchers have proposed hybrid architectures, embedding both deep learning and logical reasoning components (Santoro et al., 2017; Rocktäschel & Riedel, 2017; Manhaeve et al., 2018). Probabilistic logic reasoning has been identified as important component for explainable or interpretable machine learning, due to its ability to incorporate knowledge graphs (Arrieta et al., 2020). Combination of deep learning and logic reasoning programming have been implemented in interpretable computer vision tasks, among others (Bennetot et al., 2019; Oldenhof et al., 2023). Methods like DeepProbLog Manhaeve et al. (2018) offer an innovative approach for learning interpretable, discrete structures, such as checklists and decision trees. We study specific instantiations of these methods that are driven by real-world applications.

## 3 Background

**Problem statement:** We consider a supervised learning problem where we have access to $N$ input data points $\mathbf{x}_i \in \mathcal{X}$ and corresponding binary labels $y_i \in \{0, 1\}$. Each input data point consists of a collection of $K$ data modalities: $\mathbf{x}_i = \{\mathbf{x}_i^1, \mathbf{x}_i^2, \ldots, \mathbf{x}_i^K\}$. Each data modality can either be continuous ($\mathbf{x}_i^k \in \mathbb{R}^{d_k}$) or binary ($\mathbf{x}_i^k \in \{0, 1\}^{d_k}$). Categorical data are assumed to be represented in expanded binary format. We set $d$ as the overall dimension of $\mathbf{x}_i$. That is, $d = \sum_{k=1}^K d_k$. The $N$ input data points and labels are aggregated in a data structure $\mathbf{X}$ and a vector $\mathbf{y}$ respectively.

Our objective is to learn an interpretable decision function $f : \mathcal{X} \to \{0, 1\}$ from some hypothesis class $\mathcal{F}$ that minimizes some error criterion $d_{err}$ between the predicted and the true label. The optimal function $f^*$ then is: $f^* = \arg\min_{f \in \mathcal{F}} \mathbb{E}_{\mathbf{x}, \mathbf{y} \sim \mathcal{D}}[d_{err}(f(\mathbf{x}), \mathbf{y})]$, where $\mathcal{D}$ stands for the observational data distribution. We limit the search space of decision functions $\mathcal{F}$ to the set of predictive checklists, which are defined below.

**Predictive checklists:** Generally, we define a predictive checklist as a linear classifier applying on a list of $M$ binary concepts $\mathbf{c}_i \in \{0, 1\}^M$. A checklist will predict a data point, represented by $M$ concepts $\mathbf{c}_i = \{c_i^1, \ldots, c_i^M\}$, as positive if the number of concepts with $c_i^m = 1$ is larger or equal to a threshold $T$. That is, given a data point with concepts $\mathbf{c}_i$, the predicted label of a checklist with threshold $T$ is expressed as:

$$\hat{y}_i = \begin{cases} 1 & \text{if} \quad \sum_{m=1}^M c_i^m \geq T \\ 0 & \text{otherwise} \end{cases} \tag{1}$$

The only parameter of a checklist is the threshold $T$. Nevertheless, the complexity lies in the definition of the list of concepts that will be given as input to the checklist. This step can be defined as mapping $\psi$ that produces the binary concepts from the input data: $\mathbf{c}_i = \psi(\mathbf{x}_i)$. Existing approaches for learning checklists from data differ by their mapping $\psi$. Zhang et al. (2021) assume that the input data is already binary. In this case, the mapping is a binary matrix $\Psi_M \in \{0, 1\}^{M \times k}$ such that $\Psi_M \mathbf{1}_k = \mathbf{1}_M$, where $\mathbf{1}_k$ is a column vector of ones[1]. One then computes $\mathbf{c}_i$ as $\mathbf{c}_i = \Psi_M \mathbf{x}_i$. The element of $\Psi_M$ as well as the number of concepts $M$ (hence the dimension of the matrix) are learnable parameters.

Previous approaches (Makhija et al., 2022) relax the binary input data assumption by allowing for the creation of binary concepts from continuous data through thresholding. Writing $\mathbf{x}_i^b$ and $\mathbf{x}_i^c$ for the binary and real parts of the input data respectively, the concept creation mechanism transforms the real data to binary with thresholding and then uses the same matrix $\Psi_M$. We have $\mathbf{c}_i = \Psi_M[\mathbf{x}_i^b, \text{sign}(\mathbf{x}_i^c - \mathbf{t}_i)]$, where $[\cdot, \cdot]$ is the concatenation operator, $\mathbf{t}_i$ is a vector of thresholds, $\text{sign}(\cdot)$ is an element-wise function that returns 1 is the element is positive and 0 otherwise. In this formulation one learns the number of concepts $M$, the binary matrix $\Psi_M$ as well as the thresholds values $\mathbf{t}_i$.

**Probabilistic logic programming:** Probabilistic logical reasoning is a knowledge representation approach that involves the use of probabilities to encode uncertainty in knowledge. This is encoded in a probabilistic logical program (PLP) $\mathcal{P}$ connected by a set of $N$ probabilistic facts $U = \{U_1, ..., U_N\}$ and $M$ logical rules $F = \{f_1, ...f_M\}$. PLP enables inference on knowledge graphs $\mathcal{P}$ by calculating the probability of a query. This query is executed by summing over the probabilities of different "worlds" $w = u_1, ..., u_N$ (i.e., individual realizations of the set of probabilistic facts) that are compatible with the query $q$. The probability of a query $q$ in a program $\mathcal{P}$ can be inferred as $P_{\mathcal{P}}(q) = \sum_w P(w) \cdot \mathbb{I}[F(w) \equiv q]$, where $F(w) \equiv q$ indicates that the

---

[1]This corresponds effectively to every row of $\Psi$ summing to 1.

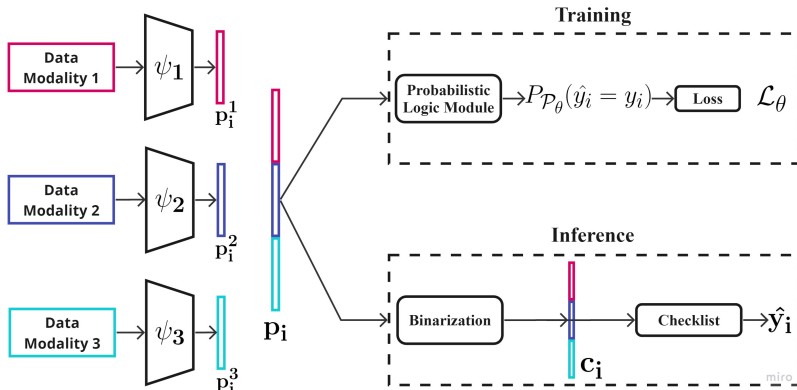

Figure 2: **Overview of our proposed ProbChecklist.** Given $K$ data modalities as the input for sample $i$, we train $K$ concept learners to obtain the vector of probabilistic concepts of each modality $\mathbf{p_i^k} \in [0,1]^{d'_k}$. Next, we concatenate into the full concepts probabilities ($\mathbf{p_i}$) for sample i. For training the concept learners, we pass $\mathbf{p_i}$ through the probabilistic logic module. At inference time, we discretize $\mathbf{p_i}$ through the thresholding parameter $\tau$ to obtain binary concepts $\mathbf{c_i}$, which are used to construct a complete predictive checklist.

propagation of the realization $w$ across the knowledge graph, according to the logical rules $F$, leads to $q$ being true. The motivation behind using a PLP is to navigate the tradeoff between discrete checklists and learnable soft concepts. Incorporating a neural network into this framework enables the generation of probabilistic facts denoted as the neural predicate $U^\theta$, where $\theta$ represents the weights. These weights can be trained to minimize a loss that depends on the probability of a query $q$: $\hat{\theta} = \arg\min_\theta \mathcal{L}(P(q \mid \theta))$. At its core, PLP provides a paradigm to manipulate probabilistic concepts according to logical rules. Considering a checklist as a specific logical rule, and concepts as probabilistic facts, probabilistic logic programming appears as a natural framework for learning checklist.We provide more details and examples of PLP in Appendix A.

## 4   ProbChecklist: learning fair and interpretable predictive checklists

### 4.1   Architecture overview

Our method first applies concept extractors, $\psi$, on each data modality. Each concept extractor outputs a list of concept probabilities for each data modality. These probabilities are then concatenated to form a vector of probabilistic concepts ($\mathbf{p_i}$) for a given data sample. This vector is dispatched to a probabilistic logic module that implements a probabilistic checklist with query $q := \mathcal{P}(\mathbf{y_i} = \hat{\mathbf{y_i}})$. We can then compute the probability of the label of each data sample and backpropagate through the whole architecture. At inference time, the checklist inference engines discretize the probabilistic checklist to provide a complete predictive checklist. A graphical depiction of the overall architecture is given in Figure 2.

### 4.2   Data modalities and concepts

Data modalities refer to the distinct sets of data that characterize specific facets of a given process. For instance, in the context of healthcare, a patient profile typically includes different clinical time series, FMRI and CT-scan images, as well as prescriptions and treatment details in text format. The division in data modalities is not rigid but reflects some underlying expert knowledge. Concepts are characteristic binary variables that are learnt separately for each modality.

### 4.3   Concept extractor

Instead of directly learning binary concepts, we extract soft concepts that we subsequently discretize. For each of the $K$ data modalities, we have a soft concept extractor $\psi_k : \mathbb{R}^{d_k} \to [0,1]^{d'_k}$ that maps the input data to a vector of probabilities $\mathbf{p}_i^k$, where $d'_k$ is the number of soft concepts to be extracted from data modality

k. If $\psi_k$ represents a neural network that is used to encode the $k^{th}$ modality, then $d'_k$ is the dimension of the output layer. Concatenating the outputs of the $K$ concept extractors results in a vector of probabilities $\mathbf{p}_i \in [0, 1]^{d'}$, with the $d'$ the total number of soft concepts $(d' = \sum_{k=1}^{K} d'_k)$.

### 4.3.1  Interpretability of the concept extractor

The concepts extractors $\psi_k$ are typically implemented with deep neural networks and are therefore not interpretable in general. To address this issue, we propose two mechanisms to improve the interpretability of the learnt concepts: (1) *focused* concept learners and (2) regularization terms that incorporate explainability in the structure of the concept learners.

**Focused concept extractor**  Focused models limit the range of features that can contribute to a concept. This increases the interpretability of each concept by explicitly narrowing down the set of features that can contribute to a given concept. Examples of focused models include LSTMs that only take specific features of the time series as input (Johnson et al., 2022), CNNs that only take a specific part of the image as input, or decision trees that only use a subset of the input features.

**Interpretability regularization**  Another approach for achieving interpretability is via regularization. For instance, TANGOS favors concepts that are obtained from distinct and sparse subsets of the input vector, avoiding overlap (Jeffares et al., 2023). As in the focused concept extractor, it allows narrowing down the set of input features that can impact a given concept. Sparsity is achieved by taking the L1-norm of the concept gradient attributions with respect to the input vector. Decorrelation between different concepts is achieved by penalizing the inner product of the gradient attributions for all pairs of concepts. More details about TANGOS and its mathematical formulation can be found in Appendix F.1.

### 4.4  Checklist learning

The checklist prediction formula in Equation 1 can be understood as a set of logical rules in a probabilistic logical program. Together with the probabilities of each concepts (*i.e.,* $d'$ probabilistic facts encoded in the vector $\mathbf{p}_i$) this represents a probabilistic logical program $\mathcal{P}_\theta$. We refer to $\theta$ as the set of learnable parameters in the probabilistic logical program.

We want to maximize the probability of a the prediction being correct. That is, we want to maximize the probability of the query $q := \hat{y}_i = y_i$,

$$\hat{\theta} = \arg\min_\theta -P_{\mathcal{P}_\theta}(\hat{y}_i = y_i) = \arg\min_\theta -\sum_w P(w) \cdot \mathbb{I}[F(w) \equiv (\hat{y}_i = y_i)] \tag{2}$$

By interpreting the probabilities $\mathbf{p}_i$ as the probability that the corresponding binary concepts are equal to 1 (*i.e.* $\mathbf{p}_i[j] = P(\mathbf{c}_i[j] = 1)$, where $[j]$ indexes the $j$-th component of the vector), we can write the probability of query $q$ as follows.

**Proposition 4.1.** *The probability of the query $\hat{y}_i = y_i$ in the predictive checklist is given by*

$$P_{\mathcal{P}_\theta}(\hat{y}_i = 1) = 1 - P_{\mathcal{P}_\theta}(\hat{y}_i = 0) = \sum_{d=T}^{d'} \sum_{\sigma \in \Sigma^d} \prod_{j=1}^{d'} (\mathbf{p}_i[j])^{\sigma(j)} (1 - \mathbf{p}_i[j])^{1-\sigma(j)} \tag{3}$$

*where $\Sigma_d$ is the set of selection functions $\sigma : [d'] \to \{0, 1\}$ such that $\sum_{j=1}^{d'} \sigma(j) = d$.*

*Proof.* Given the individual probabilities of each concept $j$ being positive, $\mathbf{p}_i[j]$, the probability that the checklist outputs exactly $d$ positive concepts for sample $i$ is given by the binomial distribution:

$$P_{\mathcal{P}_\theta}(\# \text{ true concepts} = d) = \sum_{\sigma \in \Sigma^d} \prod_{j=1}^{d'} (\mathbf{p}_i[j])^{\sigma(j)} (1 - \mathbf{p}_i[j])^{1-\sigma(j)},$$

The probability that the checklist gives a positive prediction is the probability that the number of positive concepts is larger than $T$, we thus get Equation 3. The detailed derivations are presented in Appendix A.1.  □

We use the log-likelihood as the loss function, which leads to our final loss: $\mathcal{L} = y_i \log(P_{\mathcal{P}_\theta}(\hat{y}_i = 1)) + (1 - y_i) \log(P_{\mathcal{P}_\theta}(\hat{y}_i = 0))$. The parameters $\theta$, include multiple elements: the parameters of the different soft concept extractors $(\theta_\psi)$, the number of concepts to be extracted for each data modality $d'_k$, and the checklist threshold $T$. As the soft concept extractors are typically parameterized by neural networks, optimizing $\mathcal{L}$ with respect to $\theta_\psi$ can be achieved via gradient based methods. $d'_k$ and $T$ are constrained to be integers and are thus treated as hyper-parameters in our experiments.

### 4.5 Checklist inference

ProbChecklist relies on soft concepts extraction for each data modality. Yet, at test time, a checklist operates on binary concepts. We thus binarize the predicted soft concepts by setting $\mathbf{c}_i[j] = \mathbb{I}[\mathbf{p}_i[j] > \tau]$. The thresholding parameter $\tau$ is an hyperparameter that can be tuned based on validation data. After training, we construct the final checklist by pruning the concepts that are never used in the training data (*i.e.* concepts $j$ such that $\mathbf{c}_i[j] = 0, \forall i$, are pruned). Tuning $\tau$ enables navigating the trade-off between sensitivity and specificity depending on the application.

### 4.6 Fairness regularization

We encourage fairness of the learnt checklists by equalizing the error rate across subgroups of protected variables. This is achieved by penalizing significant differences in False Positive (FPR) and False Negative Rates (FNR) for sensitive subgroups (Pessach & Shmueli, 2022). For a binary classification problem with protected attribute $S$, predicted labels $\hat{y} \in \{0, 1\}$, and actual label $y \in \{0, 1\}$, we define the differences as follows (Corbett-Davies & Goel, 2018):

$$\Delta FPR = \|P(\hat{y}_i = 1|y = 0, S = s_i) - P(\hat{y}_i = 1|y = 0, S = s_j)\|_1 \quad \forall s_i, s_j \in S \tag{4}$$

$$\Delta FNR = \|P(\hat{y}_i = 0|y = 1, S = s_i) - P(\hat{y}_i = 0|y = 1, S = s_j)\|_1 \quad \forall s_i, s_j \in S \tag{5}$$

and combine these in a fairness regularizer $\mathcal{L}_{\text{Fair}} = \lambda(\Delta FPR + \Delta FNR)$.

### 4.7 Learning checklist of decision trees

The probabilistic programming framework enables learning discrete structures, such as checklists, using gradients. Ultimately, it also allows using discrete structures as the concept extractor, like decision trees, that are more interpretable than deep learning. This leads to a combined architecture: a checklist of decision trees.

A decision tree can be represented as a logical rule. Each node is assigned a learnt concept and branches depending on whether that concept is true or false. For a tree with $L$ layers, we write $\mathbf{c}_i[j, k]$ for the learnt concept assigned to the node at layer $j$ and position $k$, for data sample $i$. If $\mathbf{c}_i[j, k] = 1$, the node branches to node $[j+1, 2k]$ and to $[j+1, 1+(k-1)*2]$ otherwise. For a tree with $L = 2$ layers and $\mathbf{p}_i[j, k] = P(c_i[j, k] = 1)$, the probability that the output of the decision tree is positive can be written as:

$$\mathcal{P}_{\text{tree}} = (1 - \mathbf{p}_i[1, 1])(\mathbf{p}_i[2, 1]) + \mathbf{p}_i[1, 1](\mathbf{p}_i[2, 2]), \tag{6}$$

This logical rule can then be combined with a checklist to form a checklist of decision trees, where the output of each tree is used as a concept in the checklist. However, the concepts used in the decision rules of the decision trees $(\mathbf{c}_i)$ still have to be learned from the data (through concepts extractors $\psi$). Simple concept extractors (*e.g.,* sigmoid function on the input data), or more complex (*e.g.,* CNN, LSTMs) can be used to balance the trade-off between interpretability and expressivity. This is in contrast with classical decision tress that can only operate on the original representation of the data (*e.g.,* a classical decision tree on an image would result in pixel-wise rules, which are likely ineffective).

While this probabilistic program is sufficient for learning decision trees, it falls short of enforcing certain desirable tree characteristics, such as being a *balanced tree*. Indeed, balanced trees are favored for their ability to provide a faithful representation of data, fostering diversity and interpretability of the learned concepts $\mathbf{c}_i$. We thus propose three regularization terms designed to facilitate the learning of balanced trees. These terms are grounded in the simple intuition that balanced trees contain distinct concepts at each node and these

concepts split the samples evenly, thereby maximizing entropy. Theses regularizations are exposed in detail in Appendix K.

Lastly, we note that checklists of trees include simple decision trees. Indeed, a checklist of trees with a single tree, leading to a checklist with a single concept and $M = 1$, is equivalent to a simple decision tree.

# 5 Experiments

We investigate the performance of ProbChecklist along multiple axes. We first compare the classification performance against a range of interpretable machine learning baselines. Second, we show how we can tune the interpretability of the learnt concepts and how we can enforce fairness constraints into ProbChecklist. Lastly, we demonstrate how our approach can be used to learn a checklist of decision tress. Complete details about the datasets, baselines used in our experiments, and hyperparameter tuning are in Appendix E and C.

**Baselines.** We compare our method against the following baselines.

*Mixed Integer Programming (MIP)*(Makhija et al., 2022). This approach allows to learn predictive checklists from continuous inputs. For images or time series, we typically apply MIP on top of an embedding obtained from a pre-trained deep learning model.

*Integer Linear Program (ILP)*(Zhang et al., 2021). ILP learns predictive checklists with Boolean inputs. We apply these to tabular data by categorizing the data using feature means as threshold.

*CNN/LSTM/BERT + Logistic Regression (LR).* This consists in using a CNN, LSTM or BERT on the input data followed by logistic regression on the combination of the last layer's embeddings of each modality.

*CNN/LSTM/BERT + Multilayer perceptron (MLP).* This is similar to the previous approach but where we apply an MLP on the combination of the last layer's embeddings of each modality.

**Datasets.** A crucial strength of our method resides in its ability to learn predictive from high dimensional input data. We briefly describe the MNIST synthetic dataset created here and defer the descriptions of other datasets (PhysioNet sepsis tabular dataset, MIMIC mortality time series dataset, Medical Abstracts TC Corpus for Neoplasm Detection) to the Appendix E.3

*Synthetic MNIST checklist.* Due to the absence of real-world datasets with ground-truth checklists, we first validate our idea on a synthetic setup created using MNIST image sequences as input and a checklist defined on digit labels. Each sample consists of a sequence of $\mathbf{K} = 4$ MNIST images (treating each image as a separate modality). We then assign a label to *each samples* according to the following ground-truth checklist. (i) Digit of **Image 1** $\in \{0, 2, 4, 6, 8\}$, (ii) **Image 2** $\in \{1, 3, 5, 7, 9\}$, (iii) **Image 3** $\in \{4, 5, 6\}$, (iv) **Image 4** $\in \{6, 7, 8, 9\}$. If at least 3 of the rules are satisfied, the label is 1, and 0 otherwise.

*PhysioNet sepsis tabular dataset.* We use the PhysioNet 2019 Early Sepsis Prediction time series dataset (Reyna et al., 2019). We transformed this dataset into tabular data by using basic summary extraction functions such as the mean, standard deviation, and last entry of the clinical time series of each patient.

*MIMIC mortality dataset.* We use the the clinical timseries data from the MIMIC III Clinical Database Johnson et al. (2016) to perform mortality prediction on the data collected at the ICU. We use hourly data of vital signs and laboratory test collected over twenty-four hours.

*Medical Abstracts TC Corpus.* We work with the clinical notes dataset designed for multi-class disease classification (Schopf et al., 2023). However, we only focus on neoplasm detection. We use medical abstracts which describe the conditions of patients. Each note is 5-7 sentences long on average.

## 5.1 Checklist performance

We evaluate the classification performance of the different models according to accuracy, precision, recall and specificity. For the checklist baselines, we also report the total number of concepts used ($M$) and the

threshold for calling a positive sample ($T$). Results are presented in table 1. Additional results and details about hyperparameter tuning are provided in Appendix E and C.

| Dataset | Model | Accuracy | Precision | Recall | Specificity | $d'_k$ | T | M |
|---|---|---|---|---|---|---|---|---|
| MNIST Checklist | CNN + MLP# | 94.72 ± 4.32 | 0.895 ± 0.1 | 0.835 ± 0.13 | 0.976 ± 0.02 | | - | - |
| | CNN + LR# | 95.04 ± 0.31 | 0.914 ± 0.01 | 0.836 ± 0.016 | **0.98 ± 0.003** | 4 | - | - |
| | pretrained CNN + MIP* | 79.56 | 0 | 0 | 1 | | 8 | 13.5 ± 0.5 |
| | **ProbChecklist** | **96.808 ± 0.24** | **0.917 ± 0.015** | **0.929 ± 0.01** | 0.978 ± 0.004 | 4 | 8.4 ± 1.2 | 16 |
| PhysioNet Tabular | Logistic Regression# | 62.555 ± 1.648 | 0.624 ± 0.0461 | 0.144 ± 0.0393 | **0.9395 ± 0.0283** | | - | - |
| | Unit Weighting* | 58.278 ± 3.580 | 0.521 ± 0.093 | 0.4386 ± 0.297 | 0.6861 ± 0.251 | 1 | 3.2 ± 1.16 | 9.6 ± 0.8 |
| | ILP mean thresholds* | 62.992 ± 0.82 | 0.544 ± 0.087 | 0.1196 ± 0.096 | 0.9326 ± 0.0623 | | 2.8 ± 0.748 | 4.4 ± 1.01 |
| | MIP Checklist* | **63.688 ± 2.437** | 0.563 ± 0.050 | **0.403 ± 0.082** | 0.7918 ± 0.06 | | 3.6 ± 0.8 | 8 ± 1.095 |
| | **ProbChecklist** | 62.579 ± 2.58 | **0.61 ± 0.076** | 0.345 ± 0.316 | 0.815 ± 0.185S | 1 | 3.6 ± 1.2 | 10 |
| MIMIC III | Unit Weighting* | 73.681 ± 0.972 | 0.469 ± 0.091 | 0.223 ± 0.206 | 0.889 ± 0.026 | | 6.1 ± 0.830 | 8.9 ± 0.627 |
| | ILP mean thresholds* | 75.492 ± 0.318 | 0.545 ± 0.028 | 0.142 ± 0.059 | 0.959 ± 0.019 | | 3.6 ± 0.894 | 3.6 ± 0.894 |
| | MIP Checklist* | 74.988 ± 0.025 | 0.232 ± 0.288 | 0.014 ± 0.017 | **0.997 ± 0.004** | 1 | 4.5 ± 2.082 | 4.5 ± 2.082 |
| | LSTM + LR# | 66.585 ± 2.19 | 0.403 ± 0.02 | **0.684 ± 0.039** | 0.66 ± 0.034 | | - | - |
| | LSTM + MLP# | 76.128 ± 0.737 | 0.446 ± 0.223 | 0.23 ± 0.132 | 0.939 ± 0.036 | | - | - |
| | LSTM + MLP# (all features) | **80.04 ± 0.598** | 0.328 ± 0.266 | 0.129 ± 0.131 | 0.962 ± 0.043 | | - | - |
| | **ProbChecklist** | 77.58 ± 0.481 | **0.642 ± 0.075** | 0.247 ± 0.032 | 0.953 ± 0.019 | 2 | 9.6 | 20 |
| Medical Abstracts Corpus | BERT + ILP* | 72.991 ± 8.06 | 0.292 ± 0.29 | 0.197 ± 0.26 | 0.879 ± 0.17 | | 1.2 ± 0.4 | 1.2 ± 0.4 |
| | BERT + MIP* | 69.32 ± 8.1 | 0.583 ± 0.14 | 0.059 ± 0.08 | 0.991 ± 0.009 | 6 | 2.5 ± 0.6 | 4 ± 0.8 |
| | BERT + LR# | 80.193 ± 0.88 | 0.790 ± 0.051 | 0.138 ± 0.065 | **0.988 ± 0.007** | | - | - |
| | BERT + MLP# | 81.782 ± 0.31 | **0.941 ± 0.04** | 0.07 ± 0.009 | 0.961 ± 0.01 | | - | - |
| | **ProbChecklist** | **83.213 ± 0.23** | 0.616 ± 0.006 | **0.623 ± 0.01** | 0.891 ± 0.003 | 6 | 3 | 6 |

Table 1: Performance results for all the models and baselines on all the datasets. We report accuracy, precision, recall as well as conciseness of the learnt checklist. We conduct a comparative analysis of ProbChecklist's performance across various metrics, acknowledging that primary metrics are often task-specific. This aspect broadens the potential applications of our method. To facilitate visualization and comparison, we plot these results in Section I of the Appendix (Figure 14). To aid the readers, we mark the non-interpretable baseline with # and existing checklist learning methods that operate only on tabular datasets with *.

**MNIST checklist dataset.** We used a simple three-layered CNN model as the concept learner for each image. In Table 1, we report the results of the baselines and ProbChecklist for $\mathbf{d'_k = 4}$ ($\mathbf{M = 16}$) on the test samples. Our method outperforms all the baselines, in terms of accuracy and recall, indicating that it identifies the minority class better than these standard approaches. The MIP failed to find solutions for some folds of the dataset and didn't generalise well on the test samples.

**Sepsis prediction from PhysioNet tabular data.** This setup is ideal for comparison with existing checklist method as they only operate on tabular dataset. In Figure 3, we visualize the checklist learnt by ProbChecklist in one of the experiments. We observe that ProbChecklist exhibits similar performance to checklist baselines.

**Mortality prediction using MIMIC mortality time series data.** To learn concepts from clinical timeseries, we use $K$ two-layered LSTMs to serve as the concept learners. We highlight our key results in Table 1. We surpass existing methods in terms of accuracy and precision by significant margin. We find that a checklist with better recall is learnt by optimizing over F1-Score instead of accuracy.

**Sepsis if 3+ items are true**

☐ **Bilirubin direct sd** $\geq 1.31$
☐ **FiO2 sd** $\geq 0.029$
☐ **FiO2 mean** $\geq 0.035$
☐ **EtCO2 sd** $\geq 0$
☐ **FiO2 last** $\geq 0.037$
☐ **Alkalinephos sd** $\geq 0$
☐ **AST sd** $\geq 0$
☐ **pH sd** $\geq 0.221$
☐ **TroponinI sd** $\geq 0$
☐ **Bilirubin direct last** $\geq 7.352$

Figure 3: Learnt checklist for PhysioNet Sepsis Prediction Task (Tabular). We report the performance result as accuracy (65.69%), precision (0.527), recall (0.755), and specificity (0.6).

**Neoplasm detection from Medical Abstracts TC Corpus.** We use a BERT model pretrained on MIMIC-III clinical notes (BioBERT) (Alsentzer et al., 2019) with frozen weights as our concept learner. We treat the entire paragraph as a single modality and feed it into BERT to obtain an M-dimensional embedding which represent soft concepts. Our checklist has a much better recall and accuracy than previous methods. Both checklist learning and deep learning methods give poor performance on the minority class.

**Sensitivity analysis.** We investigate the evolution of performance of ProbChecklist with increasing number of learnt concepts $\mathbf{d'_k}$. On Figure 4a, we show the accuracy, precision, recall, and specificity in function of the number of concepts per image on the MNIST dataset. We observe a significant improvement in performance when $\mathbf{d'_k}$ increases from **1** to **2**, and saturates after $\mathbf{d'_k} = 3$. This suggests that having learning one concept

per image is inadequate to capture all the signal in the sample and that $\mathbf{d'_k}$ is an important hyperparameter. We provide results for the sensitivity analysis on the other datasets in Appendix E.6 and additional details on hyperparameter tuning in Appendix C.

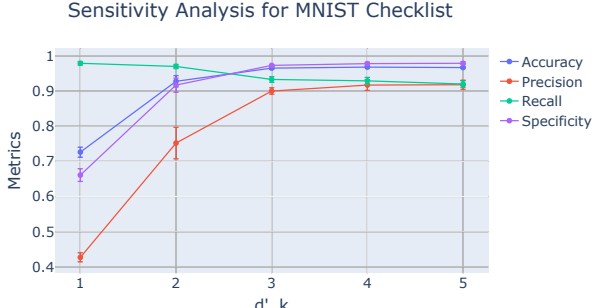

(a) Performance of ProbChecklist with varying $\mathbf{d'_k}$ on MNIST Checklist Dataset

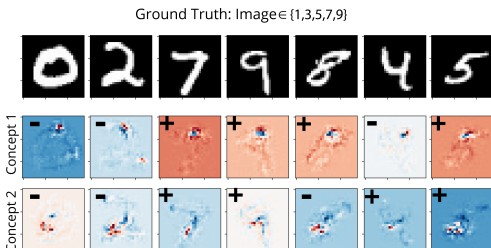

(b) We plot images and corresponding gradient attributions heat maps for seven inputs samples of the Image 2 modality of the MNIST dataset. We used a checklist with two learnable concepts per image. The intensity of red denotes the positive contribution of each pixel, whereas blue indicates the negative. If a concept predicted as true for an image, then we represent that with plus (+) sign, and with a negative sign (-) otherwise.

Figure 4: Results of ProbChecklist on MNIST Checklist Dataset: (a) Sensitivity Analysis (b) Interpretation of the concepts learnt.

## 5.2 Concepts interpretation

ProbChecklist learns interpretable concepts for tabular data by learning thresholds to binarize continuous features. However, interpreting the learnt concepts for complex modalities such as time series, images, and text is significantly more challenging. We investigate the concepts learnt from image and time series datasets with an interpretability regularization as described in Section 4.3.1. To gain insight into what patterns of signals refer to each individual concept, we examine the gradient of each concept with respect to each dimension of the input signal. Intuitively, the interpretability regularization enforces the concepts to focus on a sparse set of features of the input data.

**MNIST images.** We analyze the gradient of our checklist on individual pixels of the input images. We use a checklist with two concepts per image. On Figure 4b, we show example images of the *Image 2* of the MNIST dataset along with the gradient heat map for each learnt concept of the checklist. The ground truth concept for this image is **Image 2** $\in \{1, 3, 5, 7, 9\}$. First, we see that the 7, 9, and 5 digits are indeed the only ones for which the predicted concepts of our checklist are positive. Second, we infer from the gradient heat maps that concepts 1 and 2 focus on the image's upper half and centre region, respectively. Concept 1 is true for digits 5, 8, 9 and 7, indicating that it corresponds to a horizontal line or slight curvature in the upper half. Since Digits 0 and 2 have deeper curvature than the other images, and there is no activity in that region in the case of 4, concept 1 is false for them. concept 2 is true for images with a vertical line, including digits 9, 4, 5 and 7. Therefore, concept 2 is false for the remaining digits (0, 2, 8). The checklist outcome matches the ground truth when both concepts are true for a given image. Complementary analyses on MNIST images and MIMIC III time series is provided in Appendix F.3 and F.4. This analysis ensures interpretability at the individual sample level. As illustrated in the previous example, recognizing and comprehending these concepts at the dataset level relies on visual inspection.

**Neoplasm detection from medical abstracts.** Compared to images and time series, interpreting concepts learned from textual data is easier because its building blocks are tokens which are already human understandable. For the Neoplasm detection task, we adopt an alternative method by conducting a token frequency analysis across the entire dataset. This approach has yielded a more lucid checklist shown in Figure 1. We identified key tokens associated with positive and negative concepts (positive and negative tokens). Each concept is defined by the presence of positive words and the absence of negative words.

## 5.3 Fairness

| | Method | Female-Male | | White-Black | | Black-Others | | White-Others | |
|---|---|---|---|---|---|---|---|---|---|
| | | ΔFNR | ΔFPR | ΔFNR | ΔFPR | ΔFNR | ΔFPR | ΔFNR | ΔFPR |
| ILP mean thresholds | w/o FC | 0.038 | 0.011 | 0.029 | 0.026 | 0.152 | 0.018 | 0.182 | 0.045 |
| | FC | 0.011 | 0.001 | 0.031 | 0.008 | 0.049 | 0.016 | 0.017 | 0.0007 |
| | % ↓ | 71.053 | 90.909 | -6.897 | 69.231 | 67.763 | 11.111 | 90.659 | 98.444 |
| ProbChecklist | w/o FR | 0.127 | 0.311 | 0.04 | 0.22 | 0.02 | 0.273 | 0.02 | 0.053 |
| | FR | 0.103 | 0.089 | 0.028 | 0.016 | 0.021 | 0.008 | 0.006 | 0.008 |
| | % ↓ | 18.898 | 71.383 | 30.000 | 92.727 | -5.000 | 97.033 | 70.000 | 85.660 |

Figure 5: Improvement in fairness metrics across gender and ethnicity on MIMIC III for the mortality prediction task after adding fairness regularization. We report ΔFNR and ΔFPR for all pairs of subgroups of sensitive features and the percentage decrease (% ↓) wrt unregularized checklist.

We evaluate the fairness of ProbChecklist on the MIMIC-III Mortality Prediction task and show that we can reduce the performance disparities between sensitive attributes by incorporating fairness regularization (FR) terms, as introduced in Section 4.6. We set the sensitive features as gender $\in \{Male, Female\}$ and ethnicity $\in \{Black, White, Others\}$. Our results are displayed on Tables 5 and 12. These disparities in performance across different subpopulations are significantly reduced after fairness regularization is used. To see the effectiveness of the regularizer, we report the percentage decrease in ΔFNR and ΔFPR observed with respect to the unregularized checklist predictions for all pairs of sensitive subgroups. Similar fairness constraints (FC) can also be added to the ILP mean-thresholds baseline (Jin et al., 2022). We include a separate constraint for each pair that restricts |ΔFNR| and |ΔFPR| to be less than $\epsilon = 0.05$.

It is important to note that our approach minimizes the summation of ΔFNR and ΔFPR across all pairs of subgroups, but in the ILP we can specify a strict upper bound for each pair. Due to this, we might observe an increase in the gap for certain pairs in case of ProbChecklist, but adjusting the relative weights of these terms in the loss equation helps in achieving optimal performance. Although ProbChecklist had higher initial FNR/FPR values, the regularizer effectively reduces them to be comparable to those of ILP, particularly for the ethnicity pairs.

## 5.4 Checklist of trees

In this section, we demonstrate the flexibility of our ProbChecklist approach for learning other interpretable forms of logical decision rules and show that we can learn checklists of decision trees, where each concept is given by a decision tree. We first demonstrate that our approach can be used to learn a single decision tree and then show an example with a checklist that comprises three decision trees.

### 5.4.1 Learning a simple decision tree

To illustrate the ability of our method to learn a simple decision tree, we created a dataset where the input consists of two digits, $D1$ and $D2$, both between 0 and 9, and generated a label according to the decision tree shown in Figure 6. We used soft concept extractors of the form $\psi(\mathbf{x}) = \sigma(\beta^T \mathbf{x})$, with $\beta$ a learnable vector with same dimension than $\mathbf{x}$, and $\sigma(\cdot)$ the sigmoid function. We note that because the first rule checks if $D1$ is even, there is no decision tree with the same number of layers that can achieve 100% accuracy on this data.

We perform a train, validation, test split and report results on the test set. Figure 6 shows the ground truth tree along with the learnt decision trees (with and without balanced tree regularization). We observe that the simple ProbChecklist strategy result in a reasonable predictive performance but also note the significant added value of the balanced tree regularization, which greatly improve the performance of the learnt tree.

### 5.4.2 Learning checklists of trees

Lastly, we integrate the decision trees into the ProbChecklist framework to construct a checklist composed of trees, i.e. each concept in the checklist is the output of a decision tree. The checklist is composed of a total of T trees, with $\tau$ the checklist thresholding parameter.

We generated a synthetic dataset by using three identical decision trees with $L = 3$ layers, as shown in Figure 7. Each sample comprises three digits ($D1$, $D2$, $D3$), between 0 and 9. Each decision tree operates on a single digit. For each sample, a label was created by combining the outcomes of all decision trees, and checking whether it exceeds $T = 2$. We used the same concept extractors as in the decision tree experiment.

The learnt checklist of decision trees is shown in Figure 7, which achieves a classification performance of 0.65 AUC-ROC on the test split. It is important to note that this task is considerably more challenging than learning checklists with binary concepts or a single decision tree. In this experiment, our aim is to identify the concepts at each node of all the trees in the checklist using only the checklist outcome as the training signal. This constitutes a significantly weaker training signal compared to the previous experiments on a single decision tree.

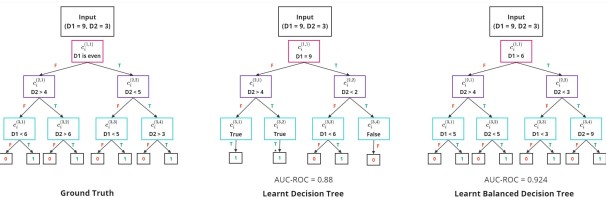

Figure 6: We illustrate both the ground truth decision tree and the learnt decision trees. The balanced tree was obtained after applying the proposed regularization scheme.

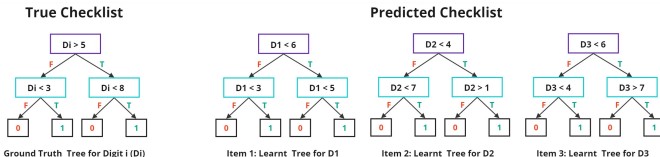

Figure 7: We illustrate both the true checklist with M = 3, T= 2 and the predicted checklist for the checklist of tree experiment. The resulting checklist had an AUC-ROC of 0.65

## 6    Discussion

**Performance of ProbChecklist.** Through these experiments, we aim to showcase that ProbChecklist surpasses existing checklist methods and achieves comparable performance to MLP (non-interpretable) methods. The switch to learnable concepts explains the improvement in accuracy over checklist methods. These concepts capture more signal than fixed summary/concept extractors used in prior works to create binarized tabular data. It's important to note that a checklist, due to its binary weights, has a strictly lower capacity and is less expressive than deep learning but possesses a more practical and interpretable structure. Despite this, it exhibits similar performance to an MLP. We also want to emphasize that ProbChecklist provides a significantly broader applicability to multimodal datasets while maintaining performance comparable to checklist baselines on tabular datasets.

**Interpretability of checklist structure and learnt concepts.** Although ProbChecklist employs a probabilistic objective for training the concept learners, the end classifier used for inference is, in fact, a discrete checklist. While this makes the classifier highly interpretable, it also shifts the focus of interpretability to the learnt concepts. We fully realize this trade-off and investigate existing techniques to maintain feature-space interpretability. For tabular datasets, ProbChecklist learns thresholds to binarize continuous features to give interpretable concepts. For time series and images, our initial results highlighted that the gradient attributions for different concepts learnt from one modality were very similar. We employ regularization terms (4.3) to enforce sparsity, avoid redundancy, and learn strongly discriminative features with high probability. We also use focused concept learners to avoid learning concepts that are functions of multiple modalities. Identifying patterns from the binarized concepts is primarily based on visual inspection and expert knowledge. Compared to images and time series, interpreting concepts from textual data is easier because text is constructed from tokens that are understandable to us. Therefore, for the medical abstract task, we identify the key tokens contributing to each concept. Lastly, it's important to note that ProbChecklist is a flexible framework, and other interpretable models can be easily incorporated as concept learners. It has been shown that Neuro-Symbolic models trained end-to-end often fall prey to reasoning shortcuts and learn concepts with unintended meanings to achieve high accuracy. This makes it harder to interpret the learnt concepts. We encourage practitioners to adopt existing methods Marconato et al. (2024) that help mitigate these issues. We offer an in-depth discussion on the interpretability of concepts for each modality in Appendix F.2.

**Extensions of ProbChecklist.** The probabilistic programming framework offers remarkable flexibility, allowing for modifications to the logical rule to accommodate various discrete structures. In our experiments, we explore decision tree learning and introduce a checklist of decision trees. Additionally, we extend our approach to learning checklists with learnable integer weights (see Appendix H). With this extension, we can assign a weight to each item, indicating its relative importance.

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

# Appendix

# A  Probabilistic Logic Programming

## A.1  Derivations for the loss function

Given the individual probabilities of each concept $j$ being positive, $\mathbf{p}_i[j]$, the probability that the checklist outputs exactly $d$ positive concepts for sample $i$ is given by the binomial distribution:

$$P_{\mathcal{P}_\theta}(\# \text{ true concepts} = d) = \sum_{\sigma \in \Sigma^d} \prod_{j=1}^{d'} (\mathbf{p}_i[j])^{\sigma(j)} (1 - \mathbf{p}_i[j])^{1-\sigma(j)},$$

where $\Sigma_d$ is the set of selection functions $\sigma : [d'] \to \{0, 1\}$ such that $\sum_{j=1}^{d'} \sigma(j) = d$. The probability that the checklist gives a positive prediction is the probability that the number of positive concepts is larger than $T$, we thus have

$$P_{\mathcal{P}_\theta}(\hat{y}_i = 1) = \sum_{d=T}^{d'} \sum_{\sigma \in \Sigma^d} \prod_{j=1}^{d'} (\mathbf{p}_i[j])^{\sigma(j)} (1 - \mathbf{p}_i[j])^{1-\sigma(j)}.$$

## A.2  Probabilistic Logic Program Example

We describe the components of a Probabilistic Logical Program (PLP) $\mathcal{P}$ with a simple example and then relate it to our checklist learning setup. $\mathcal{P}$ consists of a set of $N$ probabilistic facts $U = \{U_1, ..., U_N\}$ and $M$ logical rules $F = \{f_1, ...f_M\}$. Let's first consider a non-checklist example, the well-known Alarm Bayesian problem [1]. This example features a neighborhood that can be subject to burglaries and earthquakes. Whenever such events happen, an alarm goes off in each house and people at home make a call to the emergency station.

In this example, the probability of a burglary occurring in the neighborhood is 0.1, and the probability of an earthquake happening in that area is 0.2. Now, let's say there are two individuals, Mary and John, who live in this locality. The probability that Mary will be home is 0.4, and the probability that John will be home is 0.5.

The probabilistic facts (N=4) in this setting are:

- $U_1 : P(\text{earthquake}) = 0.2$

- $U_2 : P(\text{burglary}) = 0.1$

- $U_3 : P(\text{at\_home(John)}) = 0.5$

- $U_4 : P(\text{at\_home(Mary)}) = 0.4$

This knowledge of the world can be easily encoded in Probabilistic Logic Programs (PLPs) using these rules ($M = 3$):

- $f_1$ : alarm :- burglary (a burglary will always trigger an alarm)

- $f_1$ : alarm :- earthquake. (an earthquake will always trigger an alarm)

- $f_3$ : calls(X) :- alarm, at_home(X). (if there is an alarm, and X is at home, then X calls the station)

Probabilistic programming allows to programmatically encode the knowledge of such problems and compute the probability of different queries. A possible query $q$ for this problem could be: "What is the probability that Mary makes the call?"

Or, $q = P(\text{calls}(\text{Mary}))$.

While different implementations of probabilistic programming differ in their way to solve this problem, the simplest possible is to list all possible worlds that agree with the query and summing their probabilities. In this example, all possible worlds are given by the combinations of possible probabilistic facts.

For instance, $w_1 = $ *earthquake, not burglary, not at_home(John), at_home(Mary)* is a possible world. Furthermore, $w_1$ agrees with the observation *call(Mary)* because there is an earthquake Mary is at home, so Mary will call the station. The probability of $w_1$ is also $P(earthquake) * (1 - P(burglary)) * (1 - P(John)) * P(Mary) = 0.2 * 0.9 * 0.5 * 0.4 = 0.036$. The final answer would sum the probabilities for all such worlds that agree with the query : *(earthquake, burglary, John, Mary), (not earthquake, burglary, John, Mary), (earthquake, not burglary, John, Mary), (earthquake, burglary, not John, Mary), (not earthquake, burglary, not John, Mary), (earthquake, not burglary, not John, Mary).*

### A.3 Probabilistic Logic Program for Learning Checklists

Our checklist learning setup is analogous but provides a different set of rules that combine the original probabilistic facts (concepts) into the final prediction.

Suppose our dataset comprises $n$ points, and we are examining sample $i$ with its feature vector denoted as $x_i$. Our checklist has M concepts in total, and the probability that each concept is true for sample $i$ is represented by the vector $p_i \in [0, 1]^M$. These constitute the $M$ probabilistic facts for sample $i$ $\{U_1^{(i)}, \ldots, U_M^{(i)}\}$.

*Probabilistic Facts:*
For all $m \in [M]$:

- $U_m^{(i)} : P(\text{concept } m \text{ is true for sample } i) = p_i[m]$

The total number of probabilistic facts are $mn$. The logical rule is described by Equation 1 in the paper, which classifies a sample as positive if the number of true concepts for that sample exceeds a checklist threshold $T$.

*Logic Rules:*
For all $i \in [n]$:

- $f_i$: sample_positive(i) :- at_least_T_concepts_true (Sample i is positive if at least T concepts out of M are true for sample i)

There are $n$ logical rules in total.

Probabilistic graphical models are constructed using these probabilistic facts and logical rules, allowing PLP to perform inference on knowledge graphs by calculating the probability of a query $q$. In our context, we query "the probability that Sample i will be classified as positive." or $q = P(sample\_positive(i))$. This query is computed by summing over the probabilities of combinations that satisfy the condition $f_i$, meaning we consider all possible combinations where at least $T$ concepts are true, resulting in a positive classification.

Regarding our hybrid framework, the probabilistic facts are generated by a neural network (concept learner) with learnable weights, referred to as a neural predicate $U^\theta$, where $\theta$ represents the weights. These weights are trained to minimize a loss function based on the probability of a query $q$ and the ground truth. Given that we are dealing with binary classification problems, we employ Binary Cross Entropy as our loss function.

### A.4    Probabilistic Logic Program for Learning Decision Trees

We denote the depth of the tree with L ($> 0$) and the layers of the tree with $l \in 1, \ldots, L$. We assume a balanced binary tree structure of depth L containing $2^{l-1}$ nodes in layer $l$. The last layer contains leaf nodes that hold the model predictions. The remaining nodes represent M binary partitioning rules, where $M = \sum_{l=1}^{L-1} 2^{l-1} = 2^{L-1} - 1$. We index nodes with $(j, l)$ where $j \in 1, \ldots, 2^{l-1}$ represents the index of the node in layer $l \in 1, \ldots, L-1$. Each node takes an input vector $\mathbf{x}_i$ and outputs a boolean $c_i^{(j,l)} = \phi^{(j,l)}(\mathbf{x}_i)$. We illustrate the structure of the tree in Figure 6.

Each path of the decision tree results in a logical rule whose outcome is embedded in the value of it's leaf node. Hence, we can filter out the path which lead to a positive outcome and construct the following logical rule F, for L = 4:

$$F(\hat{y}_i = 1) := (\neg c_i^{(1,1)} \wedge \neg c_i^{(2,1)} \wedge c_i^{(3,2)}) \vee (\neg c_i^{(1,1)} \wedge c_i^{(2,1)} \wedge c_i^{(3,4)}) \vee (c_i^{(1,1)} \wedge \neg c_i^{(2,2)} \wedge c_i^{(3,6)}) \vee (c_i^{(1,1)} \wedge c_i^{(2,2)} \wedge c_i^{(3,8)}). \quad (7)$$

Let $\mathbf{p}_i[j, l] = P(c_i[j, l] = 1)$ be the probabilistic facts for all $j \in \{1, \ldots, 2^{l-1}\}$ and $l \in [L-1]$.

Now, the probability that the output of the decision tree is positive can be written as:

$$\mathcal{P}_{\text{tree}} = (1 - \mathbf{p}_i^{(1,1)})(1 - \mathbf{p}_i^{(2,1)})\mathbf{p}_i^{(3,2)} + (1 - \mathbf{p}_i^{(1,1)})\mathbf{p}_i^{(2,1)}\mathbf{p}_i^{(3,4)} + \mathbf{p}_i^{(1,1)}(1 - \mathbf{p}_i^{(2,2)})\mathbf{p}_i^{(3,6)} + \mathbf{p}_i^{(1,1)}\mathbf{p}_i^{(2,2)}\mathbf{p}_i^{(3,8)} \quad (8)$$

## B    Complexity Analysis

| Training Time (seconds) | Number of concepts | MIP Checklist | ProbChecklist |
|---|---|---|---|
| PhysioNet Tabular ($d_k' = 1$) | 10 | $667 \pm 80.087$ | $995.0 \pm 35.014$ |
| MNIST ($d_k' = 4$) | 16 | 3600 | $2773 \pm 22.91$ |

Table 2: ProbChecklist exhibits exponential complexity. To gauge the impact of this limitation, we provide training times (in seconds) for both MIP Checklist and ProbChecklist. It's crucial to highlight that while MIP Checklist performs effectively with tabular data, successfully uncovering the optimal solution, its performance is poor when applied to MNIST synthetic setup. Even when we set the runtime for Gurobi solver as 1 hour, it struggles to achieve optimal solutions for many cases. On the other hand, ProbChecklist stands out as more reliable, capable of performing end-to-end training and successfully learning the optimal solution.

Let $d$ be the number of positive concepts for sample $i$. Probabilistic Checklist Objective is defined as $P(d \geq T)$, i.e. the sample is classified as positive if $T$ or more concepts (out of $M$) are true. We discuss the space and time complexity for training and inference separately.

**Step-wise Breakdown for Training.**

- **Learning concepts probabilities using a concept learner.** The computational complexity is contingent upon the specific neural network employed. In our analysis, we utilize LSTMs, CNNs, and pre-trained BERT models with fixed weights, ensuring that computations can be performed in polynomial time.

- **Loss computation.** We need to compute the probabilistic query $\mathbf{P}(\mathbf{d} \geq \mathbf{T})$. This involves iterating over all possible $2^M$ combinations of true concepts $\forall\, d \geq T$. The probability of each combination is obtained by taking product of concept probabilities.
  Worst case time complexity: $\mathbf{O(M2^M)} \implies$ **exponential**.

We implement the time-efficient $\mathbf{O(1)}$ version of this method by caching the combination tensor in memory. In this situation, the memory cost becomes $\mathbf{O(M2^M)}$. The dimensions of tensor that stores all the combinations is $[2^M, M]$. This is an acceptable trade-off as it enables us to run experiments in reasonable time.

**Inference.** Inference involves a single pass through the concept learner. Since loss does not need to be computed, the inference time is the same for both memory- and time-efficient implementations of ProbChecklist and has the same complexity as the forward propagation of the concept learner.

**Training Details.**

We trained the time-efficient implementation with memory complexity $O(M * 2^M)$ on one NVIDIA A40 GPU with 48GB memory which can fit up to $M = 30$. Exponential complexity is primary reason for performing feature selection and limiting the modalities to 10 and learning up to 3 features per modality.

A fruitful future direction would be to study approximations to explore a smaller set of combinations.

## C    Hyperparameter Tuning

ProbChecklist framework allows experts to design user-centric checklists. The hyperparameters of ProbChecklist are concepts learned per modality $d'_k$, the parameters of the different soft concept extractors, the checklist threshold $T$ and the thresholding parameter $\tau$. Hyperparameters $d'_k$ and $T$ represent the structure/compactness of the checklist but alone aren't sufficient to garner information about the checklist's performance. Different $d'_k$ are tried for each modality in an increasing fashion to find the one that performs best. Sensitivity analysis to study the relation between $d'_k$ and performance (Figure 4a) suggests that the performance saturates after a certain point. This point can be determined experimentally for different modalities. M, total concepts in the checklist, is obtained by pruning concepts that are true for insignificant samples. Our experiments showed that pruning was not required since all the concepts were true for a significant fraction of samples. This indicates the superior quality of the concepts.

We try different values of T in the range [M/4, M/2] (total items M) to find the most performant model. However, this hyperparameter tuning doesn't contribute to the computational cost. For $d'_k$, we only only 2-3 values, and use the same value for all the features. Domain experts will be more equipped to choose these values based on their knowledge of the features. For example, if we are recording a time series feature known to stay stable (not fluctuate much), then low $d'_k$ is sufficient. The value of $d'_k$ also depends on the number of observations (most blood tests aren't performed hourly, but heart rate and oxygen saturation are monitored continuously).

### C.1    Impact of hyperparameters on performance

In Tables 7 and 10 of Appendix E.6, we present results demonstrating how the performance of ProbChecklist changes with the number of concepts learned per modality $d'_k$. We noted a significant improvement in performance when $d'_k$ increased from 1 to 2, suggesting that the samples contain complex features that a single concept cannot represent. Once all the valuable information is captured, performance reaches a saturation point and the performance plateaus with $d'_k$. This is also illustrated in Figure 4a for the MNIST Checklist dataset.
Additionally, we provide further results for the MNIST dataset to show the impact of the checklist threshold $T$ and the concept probability threshold $\tau$ hyperparameters on the performance of ProbChecklist. In Table 3, we observe a decline in performance as we deviate from the optimal value of $T$ in either direction.

## D    Conciseness of checklists over decision trees

Checklists and decision trees are both known for their interpretability in classification tasks, but there exists a tradeoff between their expressivity and interpretability. Depending on the use case, one method might be more suitable than the other. While trees grow exponentially with the number of rules, checklists only grow linearly. Yet, the increased complexity of the structure of trees makes them also more expressive. Each decision in a tree only applies to its specific lineage. In contrast, each rule in a checklist applies globally. Despite their lower expressivity, the conciseness of checklists can make them more practical and interpretable than decision trees.

| $T$ | Accuracy | Precision | Recall |
|----|----------|-----------|--------|
| 14 | 63.640 | 0.360 | 1.000 |
| 12 | 81.440 | 0.525 | 0.977 |
| 10 | 96.440 | 0.885 | 0.949 |
| 9 | 97.200 | 0.933 | 0.930 |
| 8 | 91.800 | 0.725 | 0.965 |
| 6 | 87.480 | 0.626 | 0.965 |
| 5 | 83.880 | 0.560 | 0.992 |
| 4 | 70.120 | 0.406 | 0.994 |
| 2 | 63.840 | 0.361 | 1.000 |

Table 3: Performance of ProbChecklist with varying **T** on MNIST Checklist Dataset for $d'_k = 4$ and $\tau = 0.5$

| $\tau$ | Accuracy | Precision | Recall |
|------|----------|-----------|--------|
| 0.75 | 71.29 | 0.478 | 0.911 |
| 0.6 | 96.6 | 0.892 | 0.949 |
| 0.5 | 97.2 | 0.933 | 0.930 |
| 0.4 | 63.92 | 0.362 | 1.000 |
| 0.25 | 38.13 | 0.335 | 0.926 |

Table 4: Performance of ProbChecklist with varying $\tau$ on MNIST Checklist Dataset for $d'_k = 4$ and $T = 9$

# E   Experiment Details

In this section we provide additional details about the experiments.

## E.1   Baselines

We compare the performance of our method against standard classifiers, including logistic regression (LR) and a classical multilayer perceptron (MLP). For a fair comparison, the process to obtain the **M**-dimensional concept probabilities is identical for all methods.

For the clinical datasets, we also use architectures tailored for learning checklists, namely Unit weighting, SETS checklist, Integer Linear Program (Zhang et al., 2021) with mean thresholds, and MIP from Makhija et al. (2022). These checklist-specific architectures lack the ability to process complex data modalities such as time series data, so we use features mean values for training them.

**Unit weighting** distils a pre-trained logistic regression into a checklist, as also used in Zhang et al. (2021). First, we binarise the data by using the mean feature values as thresholds. By training a simple logistic regression on the continuous data, we can determine the features with negative weights and replace them with their complements. A hyperparameter $\beta$ to discard features with weights in the range $[-\beta, \beta]$. Subsequently, we train logistic regression models on the binarized data by varying $T \in [0, M]$ (the checklist threshold) to identify the optimal value of T.

**SETS checklist** (Makhija et al., 2022) is an improvement over Unit Weighting, where mean values are directly utilized for binarization. This method aims to learn feature thresholds ($\mu$) while training a logistic regression in a temperature parameter ($\tau$) followed by unit weighting to generate the checklist.

$$\sigma(\frac{X - \mu}{\tau}) \quad \text{where, } \sigma \text{ sigmoid function.} \tag{9}$$

For **Integer Linear Program with Mean Thresholds**, we re-implement the ILP from Zhang et al. (2021) on the mean binarized data and solve the optimization problem using Gurobi. Comparing our method to this baseline provides the most relevant benchmark for evaluation.

**Mixed Integer Program** (Makhija et al., 2022) is a modification of the ILP described above and contains additional constraints to learn thresholds instead of using fixed ones. Again, we re-implement the optimization problem and solve it on Gurobi.

**BERT/LSTM/CNN + LR/MLP** employs the focused concept learners used in ProbChecklist followed by a logistic regression layer (or an MLP) on the combination of last layer's embeddings for each modality. Finally, cross-entropy loss is calculated on the predicted class, i.e. $\mathbb{P}(\hat{\mathbf{y}} = \mathbf{1})$.

### E.2  Reasons for creating MNIST synthetic Dataset

Since there aren't any real-world datasets with a ground truth checklist, we first validate our idea on a synthetic setup where the checklist is known. The key points we examine were the performance of our probabilistic checklist objective compared to standard MLP and LR architectures, the comprehensibility of the concepts learnt, and how well our approach recovers the defined rules.

The ideal dataset needed to substantiate our approach comprises different features (or modalities) and a defined rule/condition for each modality to assign the samples a ground truth label. Similar to the structure of a checklist, the sample label depends on how many rules.

These points were taken into account while designing the MNIST Synthetic Checklist.

### E.3  Datasets

**MNIST Checklist Dataset**
We generate a synthetic dataset from MNIST and define a rule set based on which the instances are classified as 0 or 1. We divide all the images in the MNIST dataset into sequences of $\mathbf{K}$ images, and each sequence forms one sample. In the experiments, we set $\mathbf{K} = \mathbf{4}$, which creates a dataset consisting of 10500 samples for training, 4500 for validation, and 2500 for testing. We start by learning $\mathbf{M}$ concepts from images directly using $\mathbf{K}$ concepts learners, which will later be used for recovering the set checklist. Concept learners would ensure that additional information beyond the labels is also captured.

We construct a ground truth checklist using the image labels to simulate a binary classification setup. A sample is assigned $\mathbf{y} = +\mathbf{1}$ if $\mathbf{T}$ out of $\mathbf{K}$ items are true. The final checklist used for experimentation is:

- **Ground Truth Checklist ($\mathbf{T} = 3$, $\mathbf{K} = 4$)**
    - ☐ **Image 1** $\in \{0, 2, 4, 6, 8\}$
    - ☐ **Image 2** $\in \{1, 3, 5, 7, 9\}$
    - ☐ **Image 3** $\in \{4, 5, 6\}$
    - ☐ **Image 4** $\in \{6, 7, 8, 9\}$

The final dataset contains 20.44% positive samples.

**PhysioNet Sepsis Tabular Dataset**
We use the PhysioNet 2019 Early Sepsis Prediction time series dataset (Reyna et al., 2019), which was collected from the ICUs of three hospitals. Hourly data of vital signs and laboratory tests are available for the thirty-four non-static features in the dataset. Additionally, four static parameters, gender, duration, age, and anomaly start point, are included. We treat the occurrence of sepsis in patients as the binary outcome variable.

The original dataset contains 32,268 patients, with only 8% sepsis patients. We create five subsets with 2200 patients, of which nearly 37% are positive. For this task, we use basic summary extraction functions such as the mean, standard deviation, and last entry of the clinical time series of each patient to tabulate the dataset. We subsequently perform feature selection and only keep the top ten informative features ($\mathbf{K} = \mathbf{10}$) based on logistic regression weights.

**PhysioNet Sepsis Time Series Data**
We use the PhysioNet 2019 Early Sepsis Prediction (Reyna et al., 2019) for this task also. The preprocessing

steps and formation of subsets are the same as described above. Instead of computing summary statistics from the clinical time series, we train different concept learners to capture the dynamics of the time series. This allows us to encapsulate additional information, such as sudden rise or fall in feature values as concepts.

The processed dataset contains 2272 training, 256 validation, 256 testing samples, and 56 time steps for each patient. We fix $K = 10$, i.e. use the top ten out of thirty-four features based on logistic regression weights. The selected features are {temperature, TroponinI, FiO2, WBC, HCO3, SaO2, Calcium, HR, Fibrinogen, AST}.

**MIMIC-III Mortality Time Series Data**
We use the clinical time series data from the MIMIC III Clinical Database (Johnson et al., 2016) to perform mortality prediction on the data collected at the intensive care facilities in Boston. We directly use the famous MIMIC-Extract (Wang et al., 2020) preprocessing and extraction pipeline to transform the raw Electronic Medical Records consisting of vital signs and laboratory records of more than 50,000 into a usable time series format. The processed dataset had a severe class imbalance with respect to the mortality prediction task. We randomly sample from the negative class to increase the percentage to 25% positive patients. Subsequently, we divide the samples into training, validation, and test subsets comprising 6912, 768, and 2048 patients. Like the PhysioNet experiments, we retain the top ten time-series features ($K = 10$). These features are {heart rate, mean blood pressure, diastolic blood pressure, oxygen saturation, respiratory rate, glucose, blood urea nitrogen, white blood cell count, temperature, creatinine}. To evaluate the proposed fairness regularizer, we introduce the one-hot encodings of two categorical features, gender and ethnicity.

**Medical Abstracts TC Corpus.** We work with the clinical notes dataset designed for multi-class disease classification (Schopf et al., 2023). However, we only focus on neoplasm detection. This subset contains 14438 total samples consisting of 11550 training samples and 2888 testing samples. Out of these, 2530 were positive in training set and 633 were positive in the testing set. To reduce class imbalance of 21.9%, the negative set was subsampled to result in an ratio of positive samples to negative samples of 35%. We use medical abstracts which describe the conditions of patients. Each note is 5-6 sentences long on average. We consider the whole text as one modality ($K = 1$).

### E.4 Sepsis Prediction using Static Tabular Data

**Data.** We use the PhysioNet 2019 Early Sepsis Prediction time series dataset (Reyna et al., 2019). We transformed this dataset into tabular data by using basic summary extraction functions such as the mean, standard deviation, and last entry of the clinical time series of each patient. We subsequently perform feature selection and only keep the top ten informative features ($K = 10$) based on logistic regression weights.

**Architecture.** Since each feature is simply a single-valued function ($x_i \in \mathbb{R}$), the concept learners are single linear layers followed by a sigmoid activation function. The remaining steps are the same as the previous task, i.e. these concept probabilities are then passed to the probabilistic logic module for $\mathbb{P}(\mathbf{d} > \mathbf{T})$ computation and loss backpropagation.

**Baselines.** We use standard baselines like MLP and LR, along with checklist-specific architectures, namely Unit weighting, SETS checklist, Integer Linear Program (Zhang et al., 2021) with mean thresholds, MIP from Makhija et al. (2022). Unit weighting distils a pre-trained logistic regression into a checklist, as also used in Zhang et al. (2021). SETS checklist (Makhija et al., 2022) consists of a modified logistic regression incorporating a temperature parameter to learn feature thresholds for binarization, this is followed by unit weighting.

**Results.** We present the results of ProbChecklist and baselines in Table 5. Our performance is slightly lower than the MIP baseline. The highest accuracy is achieved by an MLP, but that comes at the cost of lower interpretability.

### E.5 Sepsis Prediction using time series data

Instead of computing summary statistics from the clinical time series, we train different concept learners to capture the dynamics of the time series. This encapsulates additional information, such as sudden rise or fall

| Model | Accuracy | Precision | Recall | Specificity | M | T |
|---|---|---|---|---|---|---|
| Dummy Classifier | 37.226 | 0.372 | 1 | 0.628 | - | - |
| MLP Classifier | **64.962 ± 2.586** | 0.5726 ± 0.046 | 0.483 ± 0.074 | 0.76043 ± 0.0562 | - | - |
| Logistic Regression | 62.555 ± 1.648 | 0.624 ± 0.0461 | 0.144 ± 0.0393 | **0.9395 ± 0.0283** | - | - |
| Unit Weighting | 58.278 ± 3.580 | 0.521 ± 0.093 | 0.4386 ± 0.297 | 0.6861 ± 0.251 | 9.6 ± 0.8 | 3.2 ± 1.16 |
| SETS Checklist | 56.475 ± 7.876 | 0.517 ± 0.106 | **0.6639 ± 0.304** | 0.494 ± 0.3195 | 10 ± 0 | 6 ± 0.632 |
| ILP mean thresholds | 62.992 ± 0.82 | 0.544 ± 0.087 | 0.1196 ± 0.096 | 0.9326 ± 0.0623 | 4.4 ± 1.01 | 2.8 ± 0.748 |
| MIP | 63.688 ± 2.437 | 0.563 ± 0.050 | 0.403 ± 0.082 | 0.7918 ± 0.06 | 8 ± 1.095 | 3.6 ± 0.8 |
| Concepts + LR | 61.168 ± 1.45 | 0.565 ± 0.059 | 0.324± 0.15 | 0.805 ± 0.1 | - | - |
| **ProbChecklist** | 62.579 ± 2.58 | **0.61 ± 0.076** | 0.345 ± 0.316 | 0.815 ± 0.185S | 10 | 3.6 ± 1.2 |

Table 5: Performance results for Sepsis Prediction task using Tabular Data.

in feature values as concepts. Like the setup for tabular dataset, we fix **K = 10**, i.e. use the top ten features based on logistic regression weights.

We define **K** CNNs with two convolutional layers which accept one-dimensional signals as the concept learners for this task. We summarise the results for this task in Table 6. We find that ProbChecklist improves upon the baselines herein as well.

| Model | Accuracy | Precision | Recall | Specificity | $d'_k$ | M | T |
|---|---|---|---|---|---|---|---|
| Logistic Regression | 60.627 ± 1.379 | 0.4887 ± 0.106 | 0.1843 ± 0.073 | 0.8792 ± 0.048 | | - | - |
| Unit Weighting | 60.532 ± 1.567 | 0.4884 ± 0.087 | 0.1882 ± 0.102 | 0.8745 ± 0.066 | | 5 ± 0.63 | 9.2 ± 0.748 |
| ILP mean thresholds | 62.481 ± 0.426 | 0.529 ± 0.242 | 0.0529 ± 0.051 | 0.964 ± 0.031 | 1 | 3.4 ± 1.496 | 3.6 ± 1.85 |
| MIP Checklist | 60.767 ± 1.022 | 0.5117 ± 0.055 | 0.142 ± 0.05 | **0.912 ± 0.036** | | 3.5 ± 0.866 | 6.5 ± 1.5 |
| CNN + MLP | 63.465 ± 2.048 | 0.585 ± 0.05 | 0.234 ± 0.071 | 0.895 ± 0.021 | - | | - |
| CNN + MLP (all features) | **67.19 ± 0.32** | 0.526 ± 0.065 | 0.281 ± 0.041 | 0.835 ± 0.033 | | - | - |
| **ProbChecklist** | **63.671 ± 1.832** | **0.609 ± 0.115** | **0.354 ± 0.157** | 0.823 ± 0.108 | 3 | 4.4 ± 1.356 | 30 |

Table 6: Performance results for Sepsis Prediction task using Tabular Data.

### E.6 Sensitivity Analysis

We study the sensitivity of our approach to the number of learnable concepts. In particular, we compare the performance of ProbChecklistwith increasing number of learnt concepts ($d'_k$). The observed trend for both MNIST (Table 7) and PhysioNet (Table 10) datasets exhibited similarities. We noted a significant improvement in accuracy, precision, and recall when $d'_k$ increased from 1 to 2, suggesting that the samples contain complex features that a single concept cannot represent. Once all the valuable information is captured, accuracy reaches a saturation point and the performance plateaus with $d'_k$.

| $d'_k$ | Evaluation | Accuracy | Precision | Recall | Specificity | T | M |
|---|---|---|---|---|---|---|---|
| 1 | Model | 86.04 ± 0.463 | 0.814 ± 0.027 | 0.412 ± 0.021 | 0.976 ± 0.004 | 3 | 4 |
| | Checklist | 72.63 ± 1.42 | 0.427 ± 0.013 | **0.979 ± 0.005** | 0.661 ± 0.018 | | |
| 2 | Model | 96.888 ± 0.064 | 0.925 ± 0.008 | **0.922 ± 0.012** | 0.981 ± 0.003 | 5 | 8 |
| | Checklist | 92.768 ± 1.6 | 0.752 ± 0.045 | 0.97 ± 0.006 | 0.917 ± 0.02 | | |
| 3 | Model | **97.064 ± 0.187** | 0.94 ± 0.008 | 0.915 ± 0.013 | 0.985 ± 0.002 | 7 | 12 |
| | Checklist | 96.52 ± 0.33 | 0.9 ± 0.009 | 0.933 ± 0.008 | 0.973 ± 0.002 | | |
| 4 | Model | 97.04 ± 0.177 | 0.937 ± 0.005 | 0.917 ± 0.006 | 0.984 ± 0.001 | 8.4 ± 1.2 | 16 |
| | Checklist | **96.808 ± 0.24** | 0.917 ± 0.015 | 0.929 ± 0.01 | 0.978 ± 0.004 | | |
| 5 | Model | 97.032 ± 0.135 | **0.943 ± 0.005** | 0.91 ± 0.01 | **0.986 ± 0.001** | 9.4 ± 1.36 | 20 |
| | Checklist | 96.68 ± 0.269 | **0.918 ± 0.013** | 0.92 ± 0.009 | **0.979 ± 0.004** | | |

Table 7: Performance of ProbChecklist with varying $d'_k$ on MNIST Checklist Dataset

### E.7 Checklist Optimization

The checklist generation step from the learnt concepts allows users to tune between sensitivity-specificity depending on the application. The optimal checklist can be obtained by varying the threshold ($\tau$) used to binarize the learnt concepts ($\mathbf{c}_i[j] = \mathbb{I}[\mathbf{p}_i[j] > \tau]$) to optimize the desired metric (Accuracy, F1-Score, AUC-ROC) on the validation data. In Tables 9 and 8, we compare the performance of the checklist obtained by varying the optimization metric.

In the case of the MIMIC mortality prediction task, we also train our models using Binary Cross Entropy (BCE) loss and optimize over the threshold used to binarize the prediction probabilities. Next, we perform a pairwise comparison between the binary cross-entropy loss and ProbChecklist loss for each optimization metric. From the results in Table 8, it is evident that our method achieves superior performance in terms of accuracy for both metrics.

| Loss | Checklist Optimization | Accuracy | Precision | Recall | Specificity | T | M |
|---|---|---|---|---|---|---|---|
| BCE | Accuracy | 76.4±0.6 | **0.61±0.03** | 0.22±0.09 | **0.95±0.02** | - | - |
| Checklist | | **76.61±0.52** | 0.59±0.05 | **0.24±0.06** | 0.94±0.02 | 6.6±1.74 | 20 |
| BCE | F1-Score | 67.7±3.04 | 0.41±0.03 | **0.62±0.1** | 0.7±0.07 | - | - |
| Checklist | | **71.45±1.66** | **0.45±0.02** | 0.58±0.04 | **0.76±0.04** | 7.4±1.36 | 20 |
| Checklist | AUC-ROC | 34.49±4.18 | 0.27±0.01 | 0.98±0.01 | 0.13±0.06 | 4±0 | 20 |

Table 8: Results for different Checklist Optimization methods (Accuracy, F1-Score, AUC-ROC) on the MIMIC Mortality Prediction Task for $d'_k = 2$. We report results for the same architecture trained using Binary Cross Entropy (BCE) loss (instead of the proposed ProbChecklist loss) for a fair comparison.

For the PhysioNet Sepsis Prediction Task, we analyze a different aspect of our method. We study the difference in performance by changing the number of learnt concepts per modality ($d'_k$) for all the metrics.

| $d'_k$ | Checklist Optimization | Accuracy | Precision | Recall | Specificity | T | M |
|---|---|---|---|---|---|---|---|
| | Accuracy | 62.5±1.5 | 0.67±0.18 | 0.21±0.13 | 0.9±0.1 | 4±0.89 | |
| 1 | F1-Score | 49.22±9.74 | 0.44±0.06 | 0.85±0.11 | 0.27±0.22 | 3.8±0.98 | 10 |
| | AUC-ROC | 40.72±3.72 | 0.39±0.01 | 0.93±0.12 | 0.08±0.13 | 4±1.22 | |
| | Accuracy | 62.97±3.36 | 0.58±0.06 | 0.34±0.18 | 0.82±0.15 | 3±1.1 | |
| 2 | F1-Score | 38.98±1.9 | 0.39±0.02 | 0.98±0.02 | 0.01±0.01 | 2.6±0.8 | 20 |
| | AUC-ROC | 48.34±10.56 | 0.29±0.17 | 0.54±0.41 | 0.43±0.43 | 3.75±1.09 | |

Table 9: Results for different Checklist Optimization methods (Accuracy, F1-Score, AUC-ROC) on the PhysioNet Sepsis Prediction Task using timeseries. We report results for different values of $d'_k$.

### E.8 Impact of the binarization scheme

In Tables 10 and 7 we show the difference in performance between two types of evaluation schemes. The first scheme is the typical checklist, obtained by binarizing the concept probabilities and thresholding the total number of positive concepts at **T**. The second is the direct model evaluation, using the non-binarized output probability. It involves only thresholding $\mathbb{P}(\mathbf{d} > \mathbf{T})$ to obtain predictions. As expected, the performance of the checklist is slightly lower than the direct model evaluation because concept weights are binarized leading to a more coarse-grained evaluation.

| $d'_k$ | Evaluation | Accuracy | Precision | Recall | Specificity | T | M |
|---|---|---|---|---|---|---|---|
| 1 | Model | $64.105 \pm 1.69$ | $0.586 \pm 0.047$ | $0.309 \pm 0.025$ | $0.857 \pm 0.018$ | $3 \pm 0.89$ | 10 |
| | Checklist | $63.716 \pm 3.02$ | $0.613 \pm 0.12$ | $0.233 \pm 0.035$ | $0.9 \pm 0.036$ | | |
| 2 | Model | $62.656 \pm 2.377$ | $0.525 \pm 0.025$ | $0.565 \pm 0.032$ | $0.667 \pm 0.027$ | $3 \pm 1.095$ | 20 |
| | Checklist | $62.969 \pm 3.355$ | $0.582 \pm 0.061$ | $0.343 \pm 0.176$ | $0.817 \pm 0.145$ | | |
| 3 | Model | $60.781 \pm 2.09$ | $0.503 \pm 0.019$ | $0.545 \pm 0.026$ | $0.648 \pm 0.019$ | $4.4 \pm 1.356$ | 30 |
| | Checklist | $63.671 \pm 1.832$ | $0.609 \pm 0.115$ | $0.354 \pm 0.157$ | $0.823 \pm 0.108$ | | |

Table 10: Evaluation of the binarization scheme for sepsis prediction task using time series data.

## F Concepts Interpretation

### F.1 TANGOS Regularization

This section provides a detailed analysis of the learnt concepts using TANGOS regularization outlined in Section 4.3.

TANGOS regularization assists in quantifying the contribution of each input dimension to a particular concept. We examine the gradient of each concept obtained from the concept extractors with respect to the input signal. The TANGOS loss consists of two components: The first component enforces sparsity, emphasizing a concentrated subset of the input vector for each concept. The second component promotes uniqueness, minimizing the overlap between the input subsets from which each concept is derived. Sparsity is achieved by taking the L1-norm of the concept gradient attributions with respect to the input vector. To promote decorrelation of signal learned in each concept, the loss is augmented by incorporating the inner product of the gradient attributions for all pairs of concepts.

The degree of interpretability of the concepts can be varied by changing the regularization weights in the TANGOS loss equation 10. For a stronger regularization scheme (i.e. higher regularization weights), the gradient attributions are disentangled and sparse.

$$\mathcal{L}_{TANGOS} = \lambda_{sparsity}\mathcal{L}_{sparsity} + \lambda_{correlation}\mathcal{L}_{correlation} \tag{10}$$

$$= \frac{\lambda_{sparsity}}{N} \sum_{n=1}^{N} \frac{1}{d'_k} \sum_{j=1}^{d'_k} ||\frac{\partial p_j(x)}{\partial x_i}||_1 \tag{11}$$

$$+ \frac{\lambda_{correlation}}{N} \sum_{n=1}^{N} \frac{1}{d'_k C_2} \sum_{j=2}^{d'_k} \sum_{l=1}^{d'_k-1} \frac{\langle a^j(x_i), a^l(x_i) \rangle}{||a^j(x_i)||_2, ||a^l(x_i)||_2} \tag{12}$$

where, $a^j(x_i) = \frac{\partial p_j(x)}{\partial x_i}$ represents the attribution of $j^{th}$ concept with respect to the $i^{th}$ patient $(x_i)$.

To supplement convergence and enhance the stability of our models, we resort to annealing techniques to gradually increase the weights of these regularization terms.

### F.2 Discussion on the Interpretability of learnt concepts

Identifying patterns from the binarized concepts is largely based on visual inspection. To aid our analysis, we use gradient attribution of the concepts with respect to the input to identify parts that contribute to each concept.

We discuss the interpretability of the concepts for each modality separately:

- **Continuous Tabular Dataset (PhysioNet Sepsis Tabular):** Our method works effectively on continuous tabular data where we know what each attribute represents. ProbChecklist learns the

thresholds to binarize these continuous features to give the concepts. These concepts are inherently interpretable. All existing methods are designed to operate on continuous or categorical tabular datasets only. As such, our method is already novel. Nevertheless, we investigated the ability of our architecture to handle more complex data modalities.

- **Image and Time Series Tasks (MNIST/MIMIC):** Our initial results highlighted that the gradient attributions for the different concepts learned from one modality were very similar (plots in Section F.3). Pixel intensities alone are insufficient for automatically interpreting the concepts, giving rise to the need for visual inspection by domain experts. Therefore, we opted to visualize the gradient attributions of the concepts concerning the input. These plots aid domain experts in extracting patterns and recognizing the learned concepts.TANGOS enforces sparsity and decorrelation among concepts, thereby specializing them to specific input regions and preventing redundancy. While this represents one notion of interpretability, different applications may benefit from alternate definitions suited to their needs. One significant benefit of our approach is its adaptability to incorporate various other interpretability methods, enhancing its flexibility.

- **NLP Tasks (Medical Abstract Classification):** Compared to images and time series, interpreting concepts learned from textual data is easier because its building blocks are tokens which are already human understandable. In this setup, instead of employing TANGOS regularization, we identified words associated with positive and negative concepts (positive and negative tokens). Each concept is defined by the presence of positive words and the absence of negative words. The resulting checklist is visualized in Figure 1. We have edited the main paper to include more details about this.

### F.3 MNIST Images

We consider the experimental setup where each encoder learns two concepts per image. Figure 8 shows the gradient attribution heatmaps of Image 2 of the MNIST Dataset with respect to both concepts for four cases. The ground truth concept for this image is **Image 2** $\in \{1, 3, 5, 7, 9\}$.

We observe that there is a significant overlap among the gradient attributions when no regularisation terms were used ($\lambda_{sparsity} = 0$ and $\lambda_{correlation} = 0$), indicating redundancy in the learnt concepts (almost identical representations). However, it manages to identify odd digits correctly and performs well.

For $\lambda_{sparsity} = 10$ and $\lambda_{correlation} = 1$, the concepts correspond to simple features like curvatures and straight lines and are easy to identify. We infer from the gradient heat maps that concepts 1 and 2 focus on the image's upper half and centre region, respectively. Concept 1 is true for digits 5, 8, 9 and 7, indicating that it corresponds to a horizontal line or slight curvature in the upper half. Since Digits 0 and 2 have deeper curvature than the other images, and there is no activity in that region in the case of 4, concept 1 is false for them. concept 2 is true for images with a vertical line, including digits 9, 4, 5 and 7. Therefore, concept 2 is false for the remaining digits (0, 2, 8).

Table 11 highlights the tradeoff between interpretability of the concepts and the checklist performance. By increasing regularization weights, even though checklist accuracy decreases, the gradient attributions disentangle and become sparse. Based on our experiments, the checklist performance is more sensitive to the sparsity weight (i.e. decreases more sharply).

We extend this analysis to the setting with four learnable concepts $d'_k = 4$. Figure 9 ($\lambda_{sparsity} = 10$ and $\lambda_{correlation} = 1$) shows that each concept is focusing on different regions of the image. At least three concepts out of four are true for odd digits, whereas not more than one sample is true for negative samples. This attests to the ability of the method to learn underlying concepts correctly.

### F.4 MIMIC Time Series

For this analysis, we again fix our training setting to $d'_k = 2$. Sudden changes in slope in the clinical features were associated with switching of the sign gradient attributes. In Figure 10, we plot the time series for heart rate and corresponding gradient attributions for the first concept learnt by CNN. Maximum activity is observed between time steps 12 to 17 for all the patients. If a positive peak (or global maxima) is observed in

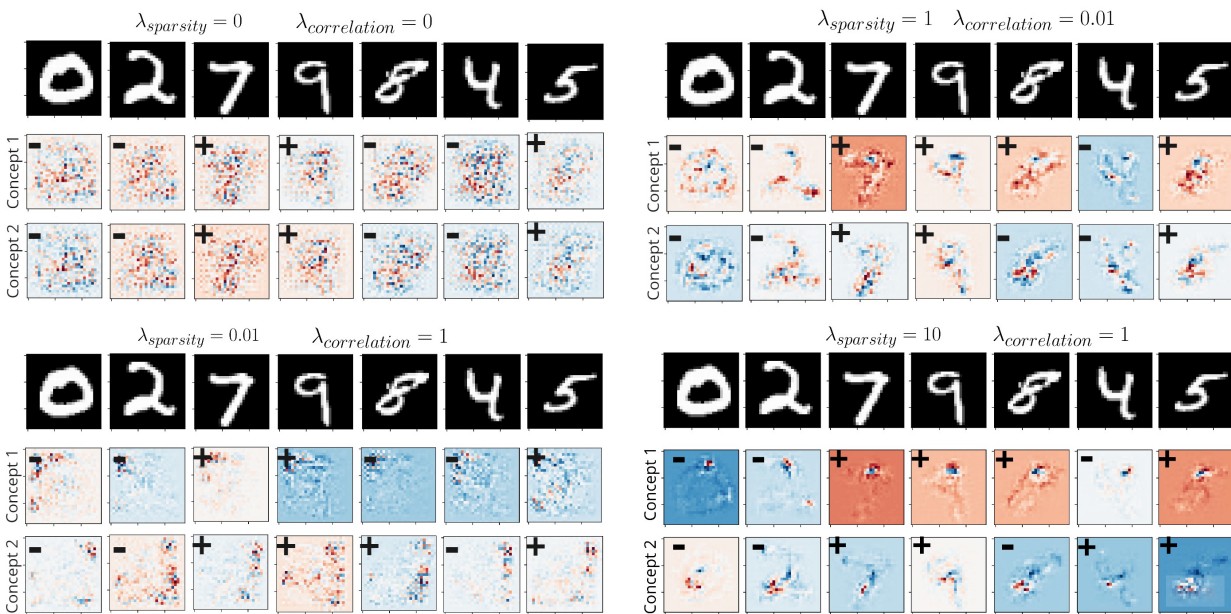

Figure 8: We plot images and corresponding gradient attributions heat maps for seven input samples of the Image 2 modality of the MNIST dataset for different combinations of sparsity and correlation regularization terms. We used a checklist with two learnable concepts per image. The intensity of red denotes the positive contribution of each pixel, whereas blue indicates the negative. If a concept is predicted as true for an image, then we represent that with a plus (+) sign and a negative (-) otherwise.

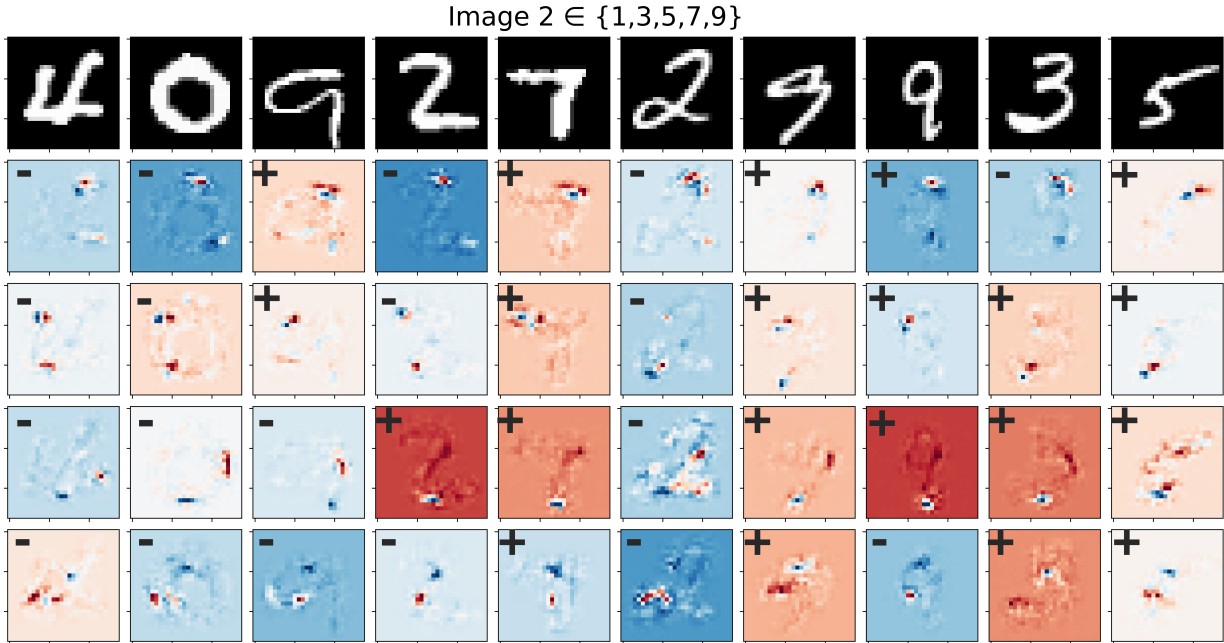

Figure 9: We plot images and corresponding gradient attributions heat maps for seven input samples of the Image 2 modality of the MNIST dataset for $\lambda_{sparsity} = 10$ and $\lambda_{correlation} = 1$. We used a checklist with four learnable concepts per image. The intensity of red denotes the positive contribution of each pixel, whereas blue indicates the negative. If a concept is predicted as true for an image, then we represent that with a plus (+) sign and a negative (-) otherwise.

| $d'_k$ | $\lambda_{\mathbf{sparsity}}$ | $\lambda_{\mathbf{correlation}}$ | Accuracy | Precision | Recall | Specificity |
|---|---|---|---|---|---|---|
| | 0 | 0 | 95.48 | 0.829 | 0.981 | 0.948 |
| 2 | 1 | 0.01 | 93.40 | 0.761 | 0.988 | 0.920 |
| | 0.01 | 1 | 92.88 | 0.745 | 0.990 | 0.913 |
| | 10 | 1 | 79.28 | 0.485 | 0.978 | 0.733 |
| | 0 | 0 | 97.20 | 0.933 | 0.930 | 0.983 |
| 4 | 1 | 0.01 | 97.04 | 0.901 | 0.961 | 0.973 |
| | 0.01 | 1 | 96.96 | 0.904 | 0.953 | 0.974 |
| | 10 | 1 | 78.88 | 0.491 | 0.844 | 0.775 |

Table 11: Performance of ProbChecklist for different combinations of sparsity ($\lambda_{sparsity}$) and correlation ($\lambda_{correlation}$) regularization weights on MNIST Checklist Dataset

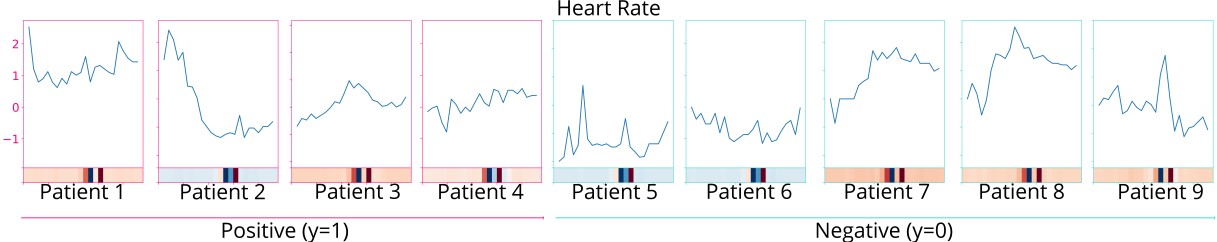

Figure 10: We plot time series for Heart Rate and corresponding gradient attributions for nine patients at $\lambda_{sparsity} = 0.1$ and $\lambda_{correlation} = 10$. The intensity of red denotes the positive contribution of that time step, whereas blue indicates the negative. The border colour of each sample encodes the ground-truth label, with pink being the positive outcome (i.e. mortality).

this period (patients 3, 6, 7, 8), then the gradient at all other time steps has a positive contribution. Another interesting observation is that if the nature of the curve is decreasing and the significant portion has negative values (Images 2, 5, 6), then the surrounding gradient attributions have a negative impact. These features are consistent irrespective of the patient outcome.

From Figure 11, we infer that gradient attributions for both the concepts are identical. In the absence of regularization terms, it was very hard to explain the concepts that were being learnt. Even though interpretability comes at the cost of performance, it helped in mapping learnt concepts to archetypal signals and patterns in the timeseries. Sudden changes in slope in the clinical features were associated with switching of the sign gradient attributes.

With the help of Figure 12, we infer the concepts being learnt when the regularization weights are $\lambda_{sparsity} = 0.1$ and $\lambda_{correlation} = 10$. The region between time steps 11 to 16 is activated for Heart Rate. The learnt concept is negative when there is a sudden increase (peak) in the values around this region as observed in Patients $4, 5, 6, 7$. This concept is true when there is slight fluctuation, or the values are steady around the high gradient region, as seen in the remaining patients. For White Blood Cell Count, our model learns two distinct concepts. The neuron gradient attributions for the first concept are high for time steps 3 to 8. This concept is positive when a local maxima is observed (an increase from the starting value and then a decrease). This pattern is visible in Patients $2, 5$. In all the other cases, a local minima is observed first.

### F.5 Medical Abstracts Corpus

Since we only work with text data, we assumed that the concepts learnt are functions of tokens in the text. To obtain interpretable summaries of the concept, we train decision trees on token occurrence and use the concept predictions as labels. We then represent each concept by the list of tokens used in the five first layers of each decision tree. We prune this representation by only keeping the unique tokens for each concept.

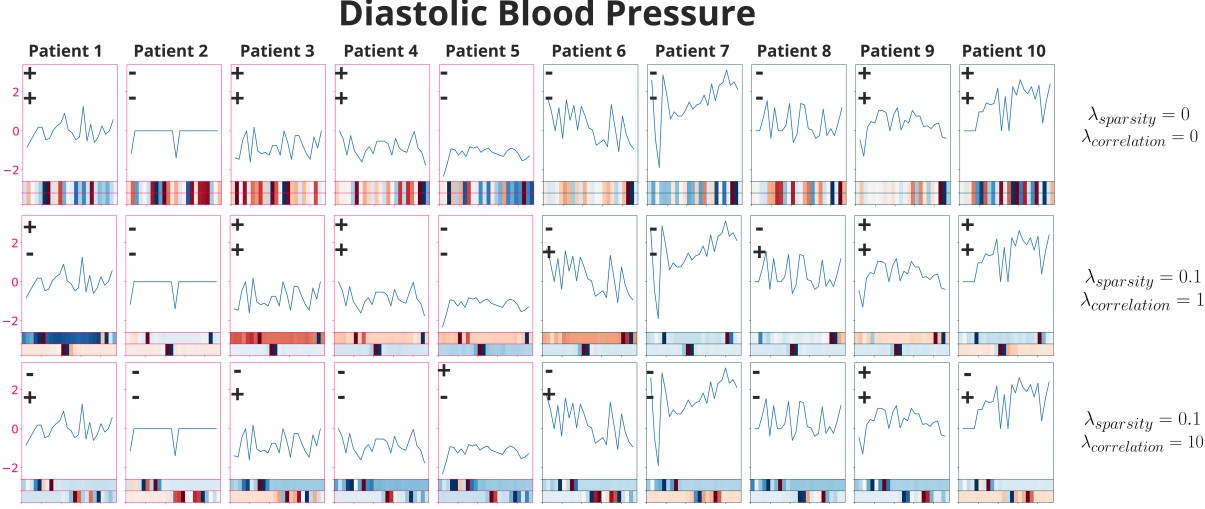

Figure 11: We plot time series for Diastolic Blood Pressure and corresponding gradient attributions for for both concepts for different combinations of regularization weights. The intensity of red denotes the positive contribution of that time step, whereas blue indicates the negative. The border colour of each sample encodes the ground-truth label, with pink being the positive outcome (i.e. mortality). If a concept is predicted as true for a patient, then we represent that with a plus (+) sign and a negative (-) otherwise.

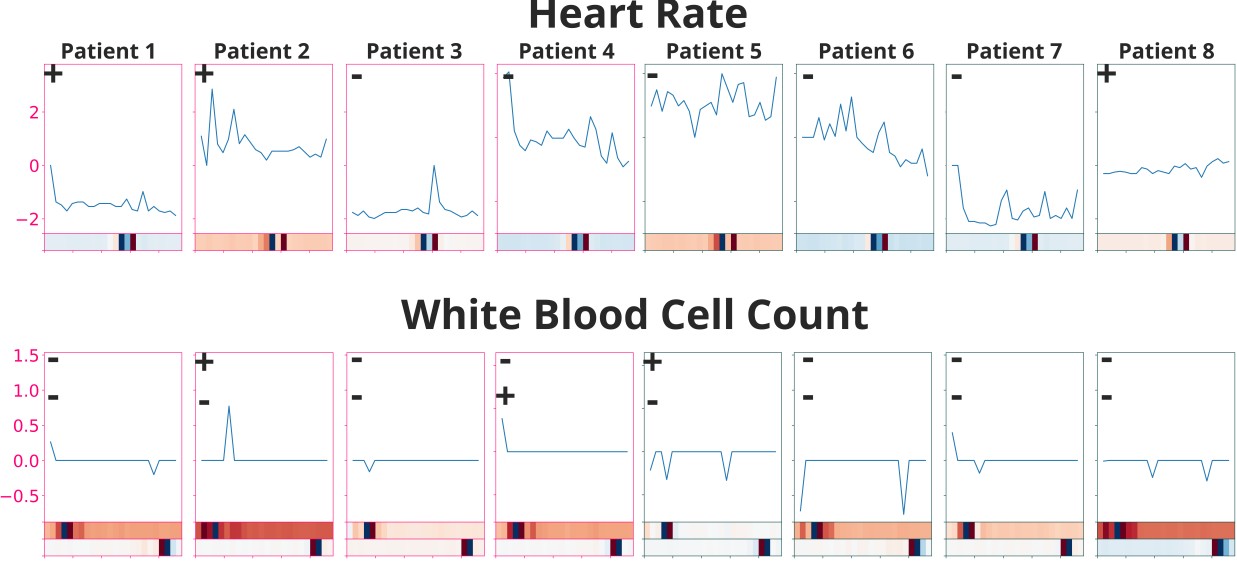

Figure 12: We plot time series for Heart Rate and White Blood Cell Count and corresponding gradient attributions for eight patients for $\lambda_{sparsity} = 0.1$ and $\lambda_{correlation} = 10$. The intensity of red denotes the positive contribution of that time step, whereas blue indicates the negative. The border colour of each sample encodes the ground-truth label, with pink being the positive outcome (i.e. mortality). If a concept is predicted as true for a patient, then we represent that with a plus (+) sign and a negative (-) otherwise.

### F.6 How domain experts can validate the concepts?

For continuous data, identifying concepts is straightforward as they are defined by the thresholds used for binarization. For instance, Figure 3 shows examples like the mean, standard deviation, and last value of medical time series data, which clinicians can verify based on domain knowledge to validate the final checklist.

In the NLP setting, we define concepts by the presence or absence of specific tokens in the text. Figure 1 contains key tokens (for neoplasm detection), derived from patient symptoms or medical reports, which can indicate the final diagnosis.

In the time series setting, we use gradient attribution analysis to highlight regions of the series that are positively or negatively activated for a concept in a patient. By repeating this for multiple patients with positive and negative outcomes, clinicians can identify trends like fluctuations, monotonicity, or constancy in the final hours, which assist in constructing checklists. A small example is provided in Appendix F.4 (Figure 10). This method relies heavily on visual inspection, but since ProbChecklist is flexible, more interpretable concept extractors can be easily integrated using existing methods.

## G   Additional Fairness Results

Additionally, we provide FNR and FPR values for each subgroup both before and after applying regularization (Table 12). This is done to ensure that the error rates of majority subgroups, where the performance was originally strong, do not increase. Notably, most minority subgroups benefit from this regularization; however, FNR increases for both the Female (minority) and Male (majority) subgroups after regularization.

| Subgroup | Before/After FR | FPR | FNR |
|---|---|---|---|
| **Female** | Before | 0.719 | 0.0989 |
|  | After | 0.1899 | 0.6975 |
| **Male** | Before | 0.4969 | 0.2931 |
|  | After | 0.127 | 0.851 |
| **White** | Before | 0.782 | 0.937 |
|  | After | 0.072 | 0.0868 |
| **Black** | Before | 0.661 | 0.824 |
|  | After | 0.0563 | 0.0597 |
| **Others** | Before | 0.724 | 0.913 |
|  | After | 0.064 | 0.0805 |

Table 12: We provide the actual values of FNR and FPR for each subgroup before and after fairness regularization is applied.

|  | Accuracy | Recall | Precision |
|---|---|---|---|
| Without Fairness Regularizer | 75.99 | 0.117 | 0.648 |
| With Fairness Regularizer | 72.412 | 0.256 | 0.553 |

Table 13: We present accuracy, precision, and recall metrics to compare the model's performance before and after applying fairness regularization. Despite observing a decrease in accuracy, the increase in recall signifies a positive development.

# H    Extension of ProbChecklist: Checklists with learnable integer weights

Checklist structure considers the weight of each item on the checklist as +1. In this section, we propose an extension to allow for integer weights larger than 1 because it has many useful applications and may be of interest to users. Before we formulate a method to learn integer weights for each concept, it is important to note that this would make it harder to interpret the checklist. More specifically, the meaning of the checklist threshold used for classification of samples would now change. Previously, it represented the minimum number of true concepts for a positive classification. Now it signifies a score - the minimum value of the weighted sum of concept probabilities for a positive classification.

Given K data modalities as the input for sample i, we train K concept learners to obtain the vector of probabilistic concepts of each modality $\mathbf{p_i^k} \in [0,1]^{d'_k}$. Next, we concatenate the full concepts probabilities ($\mathbf{p_i}$) for sample i. At this point, we can introduce a trainable weight vector $W \in [0,1]^{d'}$ (with $\sum_{i=1}^{d'} w_i = 1$) of the same dimension as $p_i$ (concept probabilities) which will capture the relative importance of the features. Element-wise product ($\odot$) of $p_i$ and W represents the weighted concept probabilities and can be denoted with $Wp_i$. This vector can be normalized by dividing each element with the maximum entry (L0 norm). While training it i For training the concept learners, we pass $Wp_i$ through the probabilistic logic module. After training, the integer weights corresponding to each concept in the checklist can be obtained by converting the W to a percentage: $W_{int}$. At inference time, we discretize $\mathbf{C_i}$ to construct a complete predictive checklist. Next, compute the score $W_{int}^T C_i$ and compare it against the checklist threshold, $M$, to classify the sample.

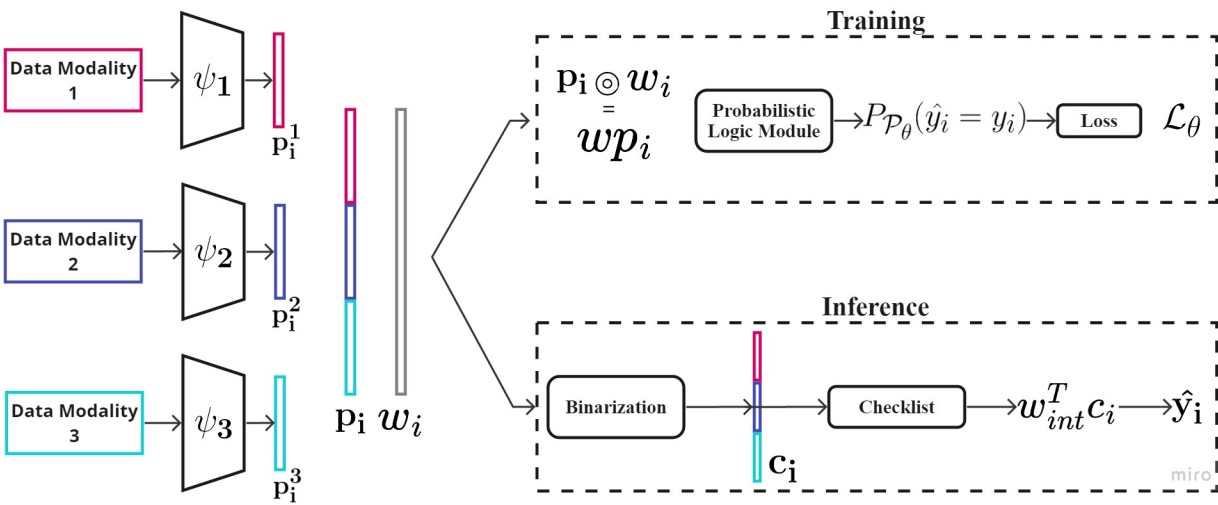

Figure 13: Checklists with learnable integer weights

# I    Plot: Performance Results of ProbChecklist

# J    Implementation

The code for all the experiments and instructions to reproduce the results are available at `https://anonymous.4open.science/r/ProbChecklist-322A/`. We have extensively used the PyTorch(Paszke et al., 2019) library in our implementations and would like to thank the authors and developers.

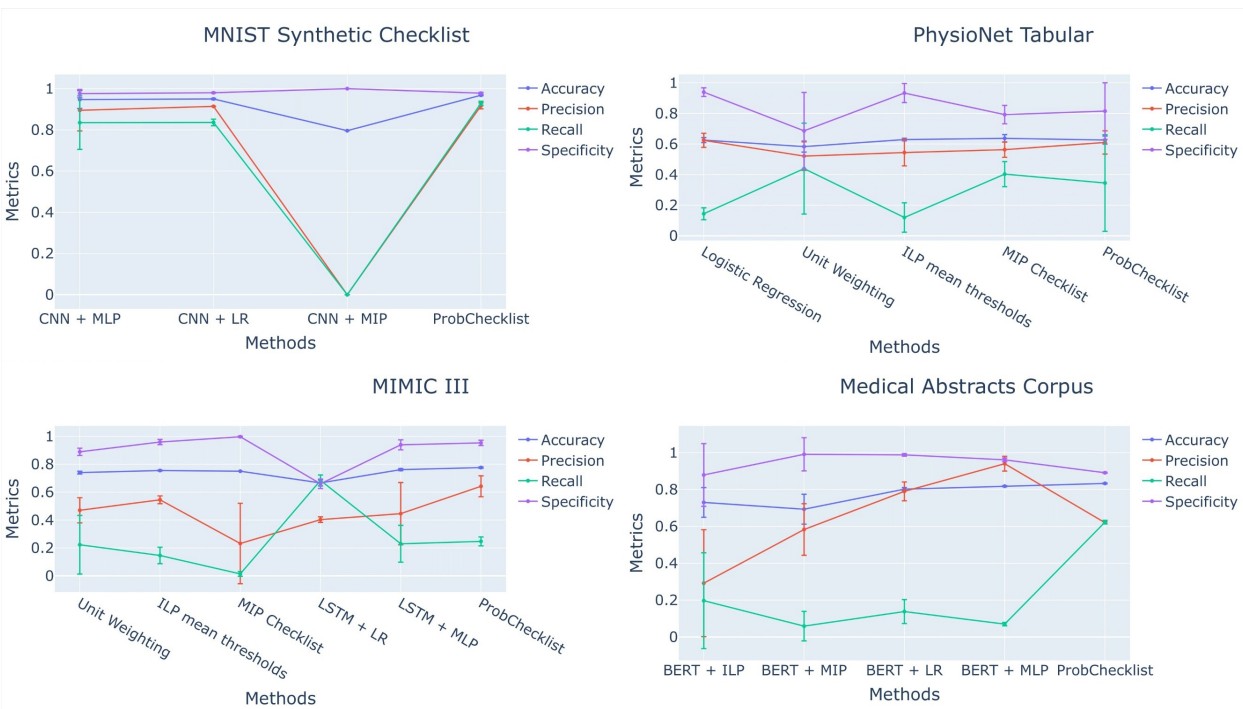

Figure 14: We plot the performance results reported in Table 1.

## K  Checklists of decision trees

### K.1  Trees as logical decision rules

We denote the depth of the tree with L $(> 0)$ and the layers of the tree with $l \in 1, \ldots, L$. We assume a balanced binary tree structure of depth L containing $2^{l-1}$ nodes in layer $l$. The last layer contains leaf nodes that hold the model predictions. The remaining nodes represent M binary partitioning rules, where $M = \sum_{l=1}^{L-1} 2^{l-1} = 2^{L-1} - 1$. We index nodes with $(j, l)$ where $j \in 1, \ldots, 2^{l-1}$ represents the index of the node in layer $l \in 1, \ldots, L-1$. Each node takes an input vector $\mathbf{x}_i$ and outputs a boolean $c_i^{(j,l)} = \phi^{(j,l)}(\mathbf{x}_i)$. We illustrate the structure of the tree in Figure 6.

Each path of the decision tree results in a logical rule whose outcome is embedded in the value of it's leaf node. Hence, we can filter out the path which lead to a positive outcome and construct the following logical rule F, for L = 4:

$$F(\hat{y}_i = 1) := (\neg c_i^{(1,1)} \wedge \neg c_i^{(2,1)} \wedge c_i^{(3,2)}) \vee (\neg c_i^{(1,1)} \wedge c_i^{(2,1)} \wedge c_i^{(3,4)}) \vee (c_i^{(1,1)} \wedge \neg c_i^{(2,2)} \wedge c_i^{(3,6)}) \vee (c_i^{(1,1)} \wedge c_i^{(2,2)} \wedge c_i^{(3,8)}).$$
(13)

When relaxing $c_i^{(j,l)}$ to be a probability between 0 and 1, we can compute the probability of positive label as

$$P_{\mathcal{P}_\theta} = P((\neg c_i^{(1,1)})(P(\neg c_i^{(2,1)})P(c_i^{(3,2)}) + P(c_i^{(2,1)})P(c_i^{(3,4)})) + P((c_i^{(1,1)})(P(\neg c_i^{(2,2)})P(c_i^{(3,6)}) + P(c_i^{(2,2)})P(c_i^{(3,8)})).$$
(14)

The node partition rules in the tree can then be learned using soft concept extractors as in Section 4 with the loss: $\mathcal{L} = y_i \log(P_{\mathcal{P}_\theta}(\hat{y}_i = 1)) + (1 - y_i) \log(P_{\mathcal{P}_\theta}(\hat{y}_i = 0))$. In our experiments, we used $M$ soft concept extractors of the form $\Psi(\mathbf{x}) = \sigma(\beta^T \mathbf{x})$, with $\beta$ a learnable vector with same dimension than $\mathbf{x}$, and $\sigma(\cdot)$ the sigmoid function. We note that although we used simple concept extractors in our experiments, they can be chosen to be arbitrarily complex. For instance, one could use a convolutional neural network on top of

images or transformers on top of text, which would produce complex decision rules. This is in contrast with classical decision tress that can only apply to the original representation of the data (*e.g.,* a classical decision tree on an image would result in pixel-wise rules, which are likely ineffective).

### K.2 Learning balanced trees

The simplest training strategy for training decision trees described above does not enforce any regularization on the produced tree, aside from the number of layers. Yet, certain tree characteristics are desirable when it comes to interpretability and diversity. In particular, balanced trees are favored for their ability to provide a faithful representation of data, fostering diversity in the learned concepts. This characteristic enhances interpretability of the concepts.

**Why do balanced trees enhance interpretability of the concepts?**

When we initially trained simple decision trees without enforcing the balanced tree constraint, we observed that some of the learned concepts were trivial (either entirely true or false). These "garbage" features led to imbalanced trees, compromising their interpretability and utility. To address this, we introduce entropy-based regularization techniques aimed at learning balanced decision trees. The intuition behind them is that each node will learn features that effectively split the samples into two significantly sized groups. This promotes interpretability by ensuring that each node's decision is meaningful rather than trivial.

Without this regularization, the tree could rely on very few concepts (or maybe even a single concept) that perfectly predict the true class, while assigning all remaining nodes are always true or false. Such a tree lacks interpretability. By enforcing balance, we ensure that most nodes contribute to learning meaningful and interpretable concepts.

**How do we learn balanced trees?**

We propose three regularization terms designed to facilitate the learning of balanced trees. These terms are grounded in the simple intuition that balanced trees contain distinct concepts at each node and these concepts split the samples evenly, thereby maximizing entropy.

Let $n^{(l,j)}$ denote the number of data points that are evaluated at $j^{th}$ node in the layer l with $N = \sum_{j=0}^{2^{l-1}} n^{(l,j)}$. Depending on the number of samples for which concepts $c_i^{(l,j)}$ is true, these $n^{(l,j)}$ points will split into two groups containing $n^{(l+1,j')}$ and $n^{(l+1,j'')}$ samples. The probabilities of the branches emerging from the split at node $c^{(l,j)}$ is the fraction of samples which are classified positive and negative. $Pr[c^{(l,j)} = 1]$ represents the fraction of samples that are classified as positive at split $c^{(l,j)}$.

$$Pr[c^{(l,j)} = 0] = \frac{n^{(l+1,j')}}{n^{(l,j)}}; \quad Pr[c^{(l,j)} = 1] = \frac{n^{(l+1,j'')}}{n^{(l,j)}}; \quad (n^{(l+1,j')} + n^{(l+1,j'')} = n^{(l,j)}) \quad (15)$$

Using these branch probabilities we can define the entropy of the split as

$$\mathsf{ENT}(l,j) = \sum_{i \in \{0,1\}} -Pr[c^{(l,j)} = i] \, log(Pr[c^{(l,j)} = i]) \quad (16)$$

The first regularization method aims to minimize the difference in entropy between the left and right subtrees generated at each split. This is achieved by computing the sum of entropy differences across all nodes where splits occur, resulting in balanced subtrees.

$$\mathcal{L}_{ENT_{Subtree}} = \sum_{l=1}^{L-1} \|\mathsf{ENT}(l+1, 2l-1) - \mathsf{ENT}(l+1, 2l)\|_1 \quad (17)$$

The second regularization term seeks to maximize the entropy of the splits in the penultimate layer of the tree (l = L-2). We sum over the entropies of all the splits in the $(L-2)$-th layer and add the negative of this

loss function.

$$\mathcal{L}_{ENT_{FinalSplit}} = \sum_{j=1}^{2^{L-1}} -\mathsf{ENT}(L-1,j) \tag{18}$$

The third regularization ensures the concepts learn from each modality are unique. To promote decorrelation of signals learned in each concept, the loss is augmented by incorporating the inner product of the concept probabilities for all pairs of concepts.

| Model | AUC-ROC | $\mathcal{L}_{\mathbf{ENT\_Subtree}}$ | $\mathcal{L}_{\mathbf{ENT\_Final\_Split}}$ |
|---|---|---|---|
| $\mathcal{P}_{Tree}$ | 0.880 | 5.502 | -2.755 |
| $+\mathcal{L}_{ENT\_Subtree}$ | 0.898 | 4.902 | -2.755 |
| $+\mathcal{L}_{ENT\_Final\_Split}$ | 0.891 | 3.673 | -3.673 |
| $+\mathcal{L}_{Correlation}$ | 0.933 | 5.306 | -3.648 |

Table 14: [Single tree] We report the results of our experiment on learning balanced decision trees using the proposed logical rule and regularization terms (see Appendix K). The first entry in the table corresponds to using only the logical rule, while the subsequent entries include the regularization terms. We observe an improvement in the AUC-ROC score as the tree becomes more balanced with the addition of the regularization terms. The decrease in the entropy-based regularization loss indicates the increased balance in the tree structure. Additionally, we present the learned decision trees in Figure 6 to corroborate these findings.

### K.3 Comparing Implementation Challenges: Decision Trees vs Checklists

In this section, we discuss the additional challenges encountered when implementing decision trees compared to checklists. Both methods, checklists and decision trees, require users to specify the number of concepts to be learned from each modality, often relying on domain knowledge. However, implementing our decision tree method is computationally intensive. In the case of decision trees, an additional difficulty arises as we also need to search over the space of possible tree structures. This introduces extra hyperparameters, such as the depth of the tree and specific node placements, which must be considered to find the optimal model. These parameters grow further when we implement a checklist of decision trees. Due to these factors, we limit our experiments with decision trees to synthetic datasets.

## L  Limitations of ProbChecklist

We have taken the first step towards learning checklists from complex modalities, whereas the existing methods are restricted to tabular data. Even though we have a mechanism to learn interpretable checklist classifiers using logical reasoning, more work is needed on the interpretability of the learnt concepts. Another drawback is the exponential memory complexity of the training. A fruitful future direction would be to study approximations to explore a smaller set of combinations of concepts. Detailed complexity analysis can be found in Appendix B. Incorporating relaxations like k-subset sampling could indeed help scale the method in practice. Drawing inspiration from papers Ahmed et al. (2023) and Wijk et al. (2024), relaxed sampling methods can be designed to approximate the top concepts and reduce memory requirements. Another idea is to explore distributed and parallel training to reduce computational bottlenecks. An active area of research focuses on parallelizing the loss Lundén et al. (2021). Drawing inspiration from similar papers, we can reduce computational requirements.

