# OpenReview forum: "Learning predictive checklists with probabilistic logic programming"
_TMLR — Rejected by TMLR_

### Review · Reviewer_veNi · 2024-08-14

**Summary Of Contributions:**

This work aims to develop predictive checklists learned from high-dimensional, continuous data. This type of predictive algorithms are motivated within the medical domain, where there is a need for automated medical checklist design, to enable interpretability/explainability of machine learning solutions.

The authors formulate predictive checklists within the framework of probabilistic logic programming, where a probabilistic algorithm learns concepts (lower dimensional embeddings) from data, to then provide a checklist over the presence or absence of a set of these concepts.

Contrary to existing approaches, the proposed algorithm extracts binary concepts from high-dimensional data according to a probabilistic checklist objective, implementable with different (even differentiable) architectures.

To enable interpretable checklist concepts, the authors propose to implement inherently interpretable concept extractors, and to use regularization penalties that enforce interpretability. Additionally, for cases where practitioners are interested in reducing performance gaps across sensitive populations, they consider the inclusion of fairness regularization constraints.

The approach is validated empirically on a synthetic (image data) and three healthcare classification tasks (tabular, time series and text), showcasing that ProbChecklist performs comparably (or better) than alternative predictive checklist approaches and interpretable machine learning baselines.

**Audience:**

Yes

**Claims And Evidence:**

Yes

**Requested Changes:**

- The authors argue that the framework is novel due to its formulation as "probabilistic logic programming".
    - I would encourage the authors to better introduce and delineate what exactly "probabilistic logic programming" entails (now split between paragraphs at the end of Section 2 and 3, with unclear connections to what is presented in Section 4).
        - How is "probabilistic logic programming" similar or disimilar to probabilistic machine learning?
    - How does the introduced terminology (probabilistic facts, logical rules and queries) translate to the context of predictive checklists?
        - what are queries in this case? How are the logical rules defined? What are the probabilistic facts in the context of predictive checklists? How is a PLP translated into learnable mapping functions?
        - How is checklist learning as presented in section 4.4. different or related to maximum likelihood estimation over probabilistic outputs?

- The authors present two variations of the proposed framework, one based on learning concepts with neural networks, another based on decision trees.
    - Can the authors elaborate and discuss on the benefits/drawbacks of each?
    - What are the experimental learnings from implementing and using one Vs the other?
        - Why not compare the approach of learning checklists of trees in the same datasets as used in Section 5.1?

- Questions on the hyperparameters of the proposed framework
    - The authors state right after Equation (1) that "the only parameter of a checklist is the threshold T".
    - However, $M$, i.e., how many concepts to learn, is also a design parameter of a checklist.
        - In Section 4.4., the authors actually acknowledge that "The parameters $\theta$, include multiple elements: the parameters of the different soft concept extractors, the number of concepts to be extracted for each data modality, and the checklist threshold T."
        - In addition, it seems that there exists a additional hyperparameter: the binary thresholding $\tau$ for the checklist.
    - I would encourage the authors to clarify and clearly state from the very beginning the hyperparameters of their proposed framework.
        - Moreover, a summary of the conclusions on the impact of these hyperparameters on the algorithm's performance (now presented in the appendix) would strengthen the quality of the work.

- Interpretability is argued to be important, yet acknowledged to be a limitation
    - For the focused concept extractor:
        - How does one limit the range of features?
        - Is this based on human/expert supervision?
    - Regularization:
        - Sparsity and decorrelation might not suffice, did the authors evaluate with experts whether learned concepts are meaningful?

- The authors argue that "Concepts are characteristic binary variables that are learnt separately for each modality", and design their method according to such assumption
    - Can the authors elaborate whether it makes sense to consider concepts learned across modalities? I.e., some medical concepts might be best captured when intersecting difference modalities.
    - How would the authors extend ProbChecklist to accommodate so?

- Other comments and suggestions
    - For clarity purposes, it might be helpful to have the exact definition of what a predictive checklist is (now in Section 3) earlier, e.g., in the introduction.
    - Given that the authors use $d$ for the dimension of $x_i$, I would suggest not using the same symbol for the error criterion $d$.
    - Figure 4a, x-axis label shows "Methods", where it should be $d_k^\prime$
    - In results for for CNN based baselines, isn't the last layer's dimensionality equivalent to $M$? Would it make sense to report it?

**Strengths And Weaknesses:**

Strengths

- ProbChecklist enables learning checklists from continuous, high-dimensional (complex) input data modalities
    - The work is of interest to the interpretability/explainability machine learning community, as well as machine learners applying these techniques within the clinical setting

- ProbChecklist does not rely in integer programming, but probabilistic machine learning.
    - Hence, it is amenable to stochastic optimization
    - ProbCheclist can be leveraged to learn a checklist of decision trees operating on concepts learnt from input data.

- Results indicate that ProbChecklist can achieve results comparable to MLP-based alternatives, yet be more "interpretable/explainable"

Weaknesses

- The proposed interpretable concept learning pipeline is, at the end, a mapping that is learned from data (typically implemented with deep neural networks), and hence, interpretability is questionable ---even if the authors proposed to use concept extractors and regularization

---

> ### Author Response · Authors · 2024-09-13
> **Author Response to Reviewer veNi (1/5)**
>
> We sincerely thank the reviewer for their constructive feedback, which has greatly contributed to improving our paper.
>
> **Strengths**
>
> The recognition of the strengths of our approach indicates that the reviewer has understood our aim to address key gaps in the checklist learning literature. Current checklist learning methods are limited, operating primarily on tabular data, whereas our proposed method expands this capability to multimodal data, narrowing that gap effectively. Additionally, the reviewer has noted the practical applicability of our method, which is an important acknowledgment of its value to the machine learning community. However, we would like to highlight one important component of ProbChecklist that may have been overlooked - the fairness regularization technique we propose. This component has demonstrated promising results and has significant practical value.
>
> **Weaknesses: Interpretability of concept learners**
>
> Our work offers more than just an interpretable method. Checklists have demonstrated advantages in rapid and clear decision-making in various fields, such as aviation, patient triaging, and surgery. We investigate whether the tedious process of designing checklists can be automated using probabilistic logical reasoning. While we do require interpretable concept extractors for this, we are fortunate to be able to rely on existing work. Although this reliance could be seen as a limitation, it's important to note that this is the first method that learns checklists from complex modalities. Beyond addressing the multimodal gap, we introduce a fairness regularization technique that has shown highly promising results with significant practical implications. Additionally, extending probabilistic logic programming to decision tree learning offers another promising avenue for future exploration.
>
> **Requested Changes**
>
> `1) Probabilistic Logic Programming Framework`
>
> `1.a)  Better introduce and delineate what exactly "probabilistic logic programming" entails. How is it similar or dissimilar to probabilistic machine learning?`
> - At its core, probabilistic logic programming provides a paradigm to manipulate probabilistic concepts according to logical rules. Considering a checklist as a specific logical rule, and concepts as probabilistic facts, probabilistic logic programming appears as a natural framework for learning checklist.
> - We have now added this to Section 3.
> - We direct the reviewer to our response to question 1.b.i, where we provide a detailed explanation of probabilistic logic programming with examples.
>
> `1.b.i) How does the introduced terminology translate to the context of predictive checklists? What are queries, logical rules, and probabilistic facts in the context of predictive checklists? How is a PLP translated into learnable mapping functions?`
> - We will attempt to clarify the components of a Probabilistic Logical Program (PLP) $\mathcal{P}$ with a simple example and then relate it to our checklist learning setup. $\mathcal{P}$ consists of a set of $N$ probabilistic facts $U = \\{U_1,...,U_N\\}$ and $M$ logical rules $F = \\{f_1,...f_M\\}$.
> - Let's first consider a non-checklist example, the well-known Alarm Bayesian problem [1]. This example features a neighborhood that can be subject to burglaries and earthquakes. Whenever such events happen, an alarm goes off in each house and people at home make a call to the emergency station.
> - In this example, the probability of a burglary occurring in the neighborhood is 0.1, and the probability of an earthquake happening in that area is 0.2. Now, let's say there are two individuals, Mary and John, who live in this locality. The probability that Mary will be home is 0.4, and the probability that John will be home is 0.5. The probabilistic facts (N=4) in this setting are:\
> U1 : P(earthquake) = 0.2 \
> U2 : P(burglary) = 0.1 \
> U3 : P(at_home(John)) = 0.5 \
> U4 : P(at_home(Mary)) = 0.4
>
> - This knowledge of the world can be easily encoded in Probabilistic Logic Programs (PLPs) using these rules (M = 3):\
> $f_1$: alarm :- burglary (a burglary will always trigger an alarm) \
> $f_2$: alarm :- earthquake. (an earthquake will always trigger an alarm) \
> $f_3$: calls(X) :- alarm, at_home(X). (if there is an alarm, and X is at home, then X calls the station)
>
> - Probabilistic programming allows to programmatically encode the knowledge of such problems and compute the probability of different queries. A possible query $q$ for this problem could be: “What is the probability that Mary makes the call?” Or, q=P(calls(Mary)).
> - While different implementations of probabilistic programming differ in their way to solve this problem, the simplest possible is to list all possible worlds that agree with the query and summing their probabilities. In this example, all possible worlds are given by the combinations of possible probabilistic facts.
>
> (We continue the response to 1.b.i in the next comment)

---

> ### Author Response · Authors · 2024-09-13
> **Author Response to Reviewer veNi (2/5)**
>
> `1.b.i Continued`\
> - For instance, w1 = earthquake, not burglary, not at_home(John), at_home(Mary) is a possible world. Furthermore, w1 agrees with the observation call(Mary) because there is an earthquake Mary is at home, so Mary will call the station. The probability of w1 is also P(earthquake) * (1-P(burglary)) * (1-P(John)) * P(Mary) = 0.2 * 0.9 * 0.5 * 0.4 = 0.036.
> - The final answer would sum the probabilities for all such worlds that agree with the query : \
> (earthquake, burglary, John, Mary), (not earthquake, burglary, John, Mary), (earthquake, not burglary, John, Mary), (earthquake, burglary, not John, Mary), (not earthquake, burglary, not John, Mary), (earthquake, not burglary, not John, Mary).
>
> **Moving to the checklist learning problem**
>
> - Our checklist learning setup is analogous but provides a different set of rules that combine the original probabilistic facts (concepts) into the final prediction. Suppose our dataset comprises $n$ points, and we are examining sample $i$ with its feature vector denoted as $x_i$. Our checklist has $M$ concepts in total, and the probability that each concept is true for sample $i$ is represented by the vector $p_i \in [0, 1]^M$. These constitute the $M$ probabilistic facts for sample $i$ $\\{U_1^{(i)}, …, U_M^{(i)}\\}$. The total number of probabilistic facts are $mn$. \
> For all $m \in [M]$ and $i \in [n]$, \
> $U_m^{(i)}$: P(concept $m$ is true for sample $i$) = $p_i$[m].
> - The logical rule is described by Equation 1 in the paper, which classifies a sample as positive if the number of true concepts for that sample exceeds a checklist threshold $T$. There are $n$ logical rules in total. \
> For all $i \in [n]$, \
> $f_i$: sample_positive(i) :- at_least_T_concepts_true (Sample $i$ is positive if at least $T$ concepts out of $M$ are true for sample $i$)
> - Probabilistic graphical models are constructed using these probabilistic facts and logical rules, allowing PLP to perform inference on knowledge graphs by calculating the probability of a query $q$. In our context, we query “the probability that Sample i will be classified as positive." or $q$ = P(sample_positive(i)).
> - This query is computed by summing over the probabilities of combinations that satisfy the condition $f_i$, meaning we consider all possible combinations where at least $T$ concepts are true, resulting in a positive classification.
> - Regarding our hybrid framework, the probabilistic facts are generated by a neural network (concept learner) with learnable weights, referred to as a neural predicate $U^\theta$, where $\theta$ represents the weights. These weights are trained to minimize a loss function based on the probability of a query $q$ and the ground truth. Given that we are dealing with binary classification problems, we employ Binary Cross Entropy as our loss function.
> - We hope this explanation addresses your concerns. These details are included in the first paragraph of Section 4.4.
>
> [1] Pearl, J., 2014. Probabilistic reasoning in intelligent systems: networks of plausible inference. Elsevier.
>
> `1.b.ii) How is checklist learning as presented in section 4.4. different or related to maximum likelihood estimation over probabilistic outputs?`
>
> - After computing the probability of the query, we proceed to minimize the loss between this probability and the ground truth to optimize the weights of our concept learners.
> - This is indeed maximum likelihood estimation (MLE) over a probabilistic output. A key point is that we use Probabilistic Programming to compute the query probability that is used in MLE.
> - Please refer to our previous response (ref), where we provide a detailed explanation of our probabilistic logical program for checklist learning.
>
> `2) The authors present two variations of the proposed framework, one based on learning concepts with neural networks, another based on decision trees.`
>
> `2.a) Can the authors elaborate and discuss on the benefits/drawbacks of each?`
>
> Checklists and decision trees are both known for their interpretability in classification tasks, but there exists a tradeoff between their expressivity and interpretability. Depending on the use case, one method might be more suitable than the other. The number of concepts in a checklist grows linearly with the number of rules, whereas in decision trees, this growth is exponential. The more complex structure of decision trees allows for greater expressivity, with decisions applied to specific lineages, while rules in a checklist apply globally. Despite their lower expressivity, the conciseness of checklists can make them more practical and interpretable. A discussion of this comparison is present in Appendix D of the paper.

---

> ### Author Response · Authors · 2024-09-13
> **Author Response to Reviewer veNi (3/5)**
>
> `2.b) What are the experimental learnings from implementing and using one Vs the other? Why not compare the approach of learning checklists of trees in the same datasets as used in Section 5.1?`
>
> - Both checklists and decision trees require users to specify the number of concepts to be learned from each modality, often relying on domain knowledge. However, implementing our decision tree method is computationally intensive. In the case of decision trees, an additional difficulty arises as we also need to search over the space of possible tree structures. This introduces extra hyperparameters, such as the depth of the tree and specific node placements, which must be considered to find the optimal model. These parameters grow further when we implement a checklist of decision trees.
> - Due to this reason, we conduct experiments for decision trees only on synthetic datasets. We have now included this in Appendix K.3
>
> `3) Hyperparameters of ProbChecklist`
>
> `3.a) The authors state right after Equation (1) that "the only parameter of a checklist is the threshold T".`
>
> In Equation 1, we define a checklist classifier. Given a fixed list of M binary concepts, samples are classified based only on the parameter T. Our method, ProbChecklist, provides a mechanism to extract these concepts from different modalities. However, it is important to note that checklists are linear classifiers defined solely by the parameter T.
>
> `3.b) However, M  i.e., how many concepts to learn, is also a design parameter of a checklist.`
>
> `3.b.i) In Section 4.4., the authors actually acknowledge that "The parameters  include multiple elements: the parameters of the different soft concept extractors, the number of concepts to be extracted for each data modality, and the checklist threshold T." `
>
> Indeed, these are the hyperparameters for the concept learners in the  ProbChecklist method.
>
> `3.b.ii) In addition, it seems that there exists a additional hyperparameter: the binary thresholding  for the checklist.`
>
> We optimize the threshold used to binarize the soft concept probabilities. We mention this in Section 4.5 - "The thresholding parameter $\tau$ is a hyperparameter that can be tuned based on validation data."
>
> `3.c) I would encourage the authors to clarify and clearly state from the very beginning the hyperparameters of their proposed framework.`
>
> We thank the reviewer for this suggestion. We have now listed all the hyperparameters for ProbChecklist together to help readers.
>
> `3.c.i) Moreover, a summary of the conclusions on the impact of these hyperparameters on the algorithm's performance would strengthen the quality of the work.`
>
> - In Table 7 of Appendix E.6 and Tables 9 and 10, we present results demonstrating how the performance of ProbChecklist changes with the number of concepts learned per modality $d_k'$. This is also illustrated in Figure 4a for the MNIST Checklist dataset.
> - Additionally, we provide further results for the MNIST dataset to show the impact of the checklist threshold T and the concept probability threshold $\tau$ hyperparameters on the performance of ProbChecklist. These results have been added to Appendix C.
> - Performance of ProbChecklist with varying $\mathbf{T}$ on MNIST Checklist Dataset for $d_k’ = 4$ and $\tau = 0.5$:
> | T  | Accuracy | Precision | Recall |
> | -- | -------- | --------- | ------ |
> | 14 | 63.640   | 0.360     | 1.000  |
> | 12 | 81.440   | 0.525     | 0.977  |
> | 10 | 96.440   | 0.885     | 0.949  |
> | 9  | 97.200   | 0.933     | 0.930  |
> | 8  | 91.800   | 0.725     | 0.965  |
> | 6  | 87.480   | 0.626     | 0.965  |
> | 5  | 83.880   | 0.560     | 0.992  |
> | 4  | 70.120   | 0.406     | 0.994  |
> | 2  | 63.840   | 0.361     | 1.000  |
>
> - Performance of ProbChecklist with varying $\tau$ on MNIST Checklist Dataset for $d_k’ = 4$ and $T = 9$:
> | $\\tau$ | Accuracy | Precision | Recall |
> | ------- | -------- | --------- | ------ |
> | 0.75    | 71.29    | 0.478     | 0.911  |
> | 0.6     | 96.6     | 0.892     | 0.949  |
> | 0.5     | 97.2     | 0.933     | 0.930  |
> | 0.4     | 63.92    | 0.362     | 1.000  |
> | 0.25    | 38.13    | 0.335     | 0.926  |
>
> `4) Interpretability is argued to be important, yet acknowledged to be a limitation`
>
> Interpretability is a major motivation of our work. However we also honestly acknowledge the difficulties in building a fully interpretable and expressive machine learning pipeline. In our case, while our end classifier (a discrete checklist) is highly interpretable, it reports some of the interpretability burden on the concept extractors. However, these concept extractors can benefit from other interpretability techniques from prior works. We spend a significant effort in the experiments assessing the interpretability of the concepts. This tradeoff is discussed in Section 6.

---

> ### Author Response · Authors · 2024-09-13
> **Author Response to Reviewer veNi (4/5)**
>
> `4.a) For the focused concept extractor: How does one limit the range of features? Is this based on human/expert supervision?`
>  - ProbChecklist allows users to provide hyperparameters $d'_k$, which denote the number of concepts learnt from modality k. If the concept learner for modality k is a neural network, then this represents its final embedding dimension. Indeed, this $d'_k$ can be determined using domain knowledge of experts.
> - In the absence of experts, different $d'_k$ are tried for each modality in an increasing fashion to find the one that performs best. We conduct sensitivity analysis to study the relation between $d'_k$ and results (Figure 4a) suggest that the performance saturates after a certain point. This point can be determined experimentally for different modalities.
> - This information is present in Appendix C (Hyperparameter Tuning) and results for the sensitivity analysis on the other datasets are in Appendix E.6
>
> `4.b) Regularization: Sparsity and decorrelation might not suffice, did the authors evaluate with experts whether learned concepts are meaningful?`
> - We employ sparsity and decorrelation regularization for both image and time series tasks. To support pattern identification based on visual inspection, we use gradient attribution of the concepts with respect to the input to identify the parts contributing to each concept. For deployment in real-world scenarios, expert analysis would be required to interpret the learnt concepts. However, this is beyond the scope of our present work.
> - In Appendix F.3, we highlight the improvements observed with these regularization terms by plotting the gradient attribution. Figure 8 clearly shows that without the proposed regularization (sparsity and decorrelation), the gradient attributions for all concepts are nearly identical, indicating that concepts are redundant.
> - We plot gradient attributions after applying regularization and compare the activated regions for different inputs to identify concepts. This is elaborated in Section 5.2 for the MNIST dataset. Similar analysis for MIMIC can be found in Appendix F.4
>
> `5) The authors argue that "Concepts are characteristic binary variables that are learnt separately for each modality", and design their method according to such assumption: Can the authors elaborate whether it makes sense to consider concepts learned across modalities? I.e., some medical concepts might be best captured when intersecting difference modalities. How would the authors extend ProbChecklist to accommodate so?`
>
> ProbChecklist is a flexible method that enables users to customize the design of concept learners according to their specific use cases. For instance, if it is more advantageous to integrate two modalities and learn a combined set of concepts, users can opt for a single concept learner and provide it with concatenated input. However, this approach may make it more challenging to interpret the learnt concepts. Therefore, we recommend using focused concept learners (one for each modality type). A significant advantage of our method is that it offers users the flexibility to tailor the architecture to suit their needs.
>
> `6) Other comments and suggestions` \
> `6.a) For clarity purposes, it might be helpful to have the exact definition of what a predictive checklist is (now in Section 3) earlier, e.g., in the introduction.` \
> We thank the reviewer for this suggestion. We have made this change for enhanced clarity.
>
> `6.b) Given that the authors use d for the dimension of x_i, I would suggest not using the same symbol for the error criterion d.` \
> This was an inadvertent oversight on our part. We have changed the notation for the error criterion.
>
> `6.c) Figure 4a, x-axis label shows "Methods", where it should be d_k’` \
> The incorrect labeling of the x-axis in Figure 4a was an inadvertent oversight on our part. We thank the reviewer for bringing this to our attention and we have corrected it accordingly.
>
> `6.d) In results for for CNN based baselines, isn't the last layer's dimensionality equivalent to $\mathcal{M}$? Would it make sense to report it?`
> - We have employed focused CNNs even for the CNN+MLP method, meaning there is one CNN for each image. The final embedding dimension for each CNN is 4. We have included this information in the experimental details (Appendix E.1).
> - We use the notation M and T to define the structure of the checklist. Using this notation to denote the dimensionality of the embedding size for a non-checklist method in Table 1 might cause confusion.
> - Additionally, we want to clarify that M differs from the final embedding size for ProbChecklist, as some of these concepts are either redundant or consistently true or false and are not included in the checklist. We have mentioned this in Section 4.5

---

> ### Author Response · Authors · 2024-09-13
> **Author Response to Reviewer veNi (5/5)**
>
> **New Revision**\
> We have incorporated your feedback in the new revision (all changes are highlighted in blue). The changes are:
> - Added the definition of a predictive checklist in the introduction.
> - Modified the notation for the error criterion in Section 3.
> - Introduced a new paragraph at the end of Section 3 to motivate the use of probabilistic logic programming for checklist learning.
> - Listed all the hyperparameters of ProbChecklist.
> - Updated Figure 4a with corrected axis labels.
> - Added Appendix C.1 to discuss the impact of hyperparameters on ProbChecklist's performance.
> - Made minor adjustments to Appendix D, comparing the advantages and disadvantages of checklist learning versus decision tree learning.
> - Provided additional details about the BERT/LSTM/CNN + LR baseline in Appendix E.1.
> - Added Appendix K.3 to highlight implementation challenges associated with our decision tree learning method.
>
> ---
> Thank you again for reviewing our paper and for your encouraging feedback!

---

### Review · Reviewer_n7Qo · 2024-09-19

**Summary Of Contributions:**

This paper learns a predictive checklist (ie, a binary classification) from high-dimensional inputs by using neural networks to extract high-level concepts, and using probabilistic reasoning to calculate the probability that more than $T$ concepts are positive. The system is learned end-to-end. The authors furthermore study the interpretability and fairness of the system, adding extra components to encourage these desirable properties.

**Audience:**

Yes

**Claims And Evidence:**

Yes

**Requested Changes:**

- (Strengthen): Although I get how the loss arises from the PLP machinery, I don't see how PLP is useful in this paper. The background section on PLP is hard to follow and very vague. It is unclear what actual PLP language the authors are using. However, the actual loss function comes from the idea that you want at least $T$ positive items - This is very easy to express without PLP, and this is in fact done in Appendix A. Then there is no need to talk about probabilistic facts, logic rules etc.
- (Critical): In case the PLP is kept, please clarify and formalise the PLP setup:
    - S3: The 'probabilistic facts' are used as atoms
    - S3: The explanation of the probability of a query (the weighted model count) is very unclear. "The propagation of the realization w across the knowledge graph, according to the rules $F$, leads to $q$ being true". What is the knowledge graph? How are the rules defined? Are those horn clauses?
    - S4.4: The goal of PLP is to create a program, but I see none in the method. What is the actual program used to encode the checklist? Eq 2 talks about some $F$ that supposedly encodes the logic program but never defines it. And for the decision tree learning?
- (Critical): Appendix A should be part of the main text. It's a simple and short argument that was critical in helping me understanding what was going on.
- (Critical): Especially if PLP is kept, the comparison to DeepProbLog (Manhaeve et al. 2018) should be more thorough, as ProbCheckList can be directly encoded in DeepProbLog.
- (Critical): The complexity analysis can be significantly improved. There are papers that use dynamic programming to compute this in polytime [1, 2, 3] for the case where the number of true concepts is exactly $d$, these need to be discussed. It should be possible to scale far beyond M=30 with these by doing this DP algorithm for everything above $T$.
- (Critical): Concepts that are trained end-to-end without direct supervision like in this paper will have reasoning shortcuts [4] (concepts that don't have their intended meaning - such as in your MNIST example). A short discussion on this limitation is important.

Smaller points:
- In Figure 1, what do \oly, \ymph etc represent?
- Sec 2: "Our method does not rely on integer programming and thus exhibits much faster computing times". I think this is quite arguable, considering that the method requires computing the loss function that, until now, cannot be scaled beyond $M=30$.
- Sec 3: I don't understand the binary mapping $\Psi_M 1_k=1_k$. Shouldn't the output be $1_M$?
- Sec 4: I got confused by the hyperparameter $d'_k$ I think it could have more emphasis in how it's defined.
- Sec 4: The distinction between $q$, $y_i$ and $\hat{y}_i$ is lost on me.
- Sec 4.3.1: The 'focused concept extractor' just seems like feature selection to me?
- Sec 4.4: Eq 2 is a lot easier to understand as it's posed in the appendix. Currently without a clear definition of $F$, it's very hard to follow.
- Sec 4.4: Proposition 4.1 is not formal enough. There is no clear statement of the assumptions or the program from which this result follows. Furthermore, the $\Sigma_d$ and $\sigma$ seem somewhat like superfluous notation for the purposes of this paper and is not used anywhere else.
- Sec 4.7: Eq 6 seems like the last term should be $p_i[2, 2]$, not $p_i[2, 4]$?
- Sec 4.7: It's not clear to me why balanced trees would 'provide a faithful representation of data, fostering diversity and interpretability of the learned concepts'
- Sec 5: I don't understand how MIP and ILP  are used. Those aren't learning algorithms, right?
- Sec 5: I think it should be highlighted that the MNIST problem can be solved with $d'_k=1$, suggesting that the algorithm isn't learning the right / interpretable concepts since $d'_k=4$ is needed.
- Sec 5.1: Table 1: I don't understand why there are error bars under $T$. I thought this was a hyperparameter?
- Sec 5.1: Table 1: Under precision - PhysioNet Tabular, ProbCheckList is bolded but LR has a higher score.
- Sec 5.1: The order in which the datasets are discussed is different from Table 1
- Sec 5.2: I understand why this is studied, but I don't think this feature attribution method relying on 'visual inspection' is a thorough and infallible way to introduce interpretability...

[1] Ahmed, Kareem, et al. "SIMPLE: A Gradient Estimator for k-Subset Sampling." The Eleventh International Conference on Learning Representations.
[2] Chen, Sean X., and Jun S. Liu. "Statistical applications of the Poisson-binomial and conditional Bernoulli distributions." Statistica Sinica (1997): 875-892.
[3] Wijk, Klas, Ricardo Vinuesa, and Hossein Azizpour. "Revisiting Score Function Estimators for $ k $-Subset Sampling." arXiv preprint arXiv:2407.16058 (2024).
[4] Marconato, Emanuele, et al. "Not all neuro-symbolic concepts are created equal: Analysis and mitigation of reasoning shortcuts." Advances in Neural Information Processing Systems 36 (2024).

**Strengths And Weaknesses:**

**Strengths**: The motivation and idea behind the approach are interesting, and this seems like a good application. I like the idea of creating an ML model that follows in pipeline a process that typical medical professionals would perform. The experiments are fairly thorough both on some toy datasets with MNIST and some medical datasets.

**Weaknesses**: I found the method section hard to follow, given that the main idea is pretty simple (where simplicity is a good thing!). I don't believe the probabilistic logic programming (PLP) machinery adds anything to the understanding and overcomplicates the exposition (see below for details). There is some related work that's missing or could be more thoroughly discussed. Finally,  interpretability of the method is quite limited, as recognised by the authors.

---

> ### Author Response · Authors · 2024-11-02
> **Author Response to Reviewer n7Qo (1/6)**
>
> We sincerely thank the reviewer for their constructive feedback, which has greatly contributed to improving our paper.
>
> **Strengths**
>
> The recognition of the strengths of our approach indicates that the reviewer has understood our aim to address key gaps in the checklist learning literature. Current checklist learning methods are limited, operating primarily on tabular data, whereas our proposed method expands this capability to multimodal data, narrowing that gap effectively. We thank the reviewer for the encouraging feedback.
>
> **Weaknesses**
> - *[Regarding the complexity of the method]*:  We believe that Probabilistic Logic Programming (PLP) is central to our approach because it provides a flexible framework to learn various discrete structures like checklists and decision trees. The PLP framework allows us to directly support the checklist of trees approach of Section 5.4. The PLP formalism allows to naturally and elegantly collect these different decision rules under a single umbrella, using logic-based reasoning for learning. Nevertheless, we understand that this comes at the cost of extra complexity of the method. To tackle this concern, we have adapted and simplified  the exposition of PLP in the paper. For better clarity, we have added a toy ProbLog program example and made comparisons with the checklist learning program in Appendix A to help understand the PLP terminology.
> - *[Regarding the interpretability of our method]* :  Although interpretable concept extractors are required for our method, we are fortunate to rely on the rich existing literature in this area. We want to emphasize that our primary objective and motivation was to study the feasibility of checklist learning from complex data modalities. We therefore relied on existing work for providing interpretable concept learners.e  We have clarified this distinction  by adding a detailed discussion on the interpretability of the learned concepts.
> - *[About missing related work]*:  We have  incorporated the related work suggested by the reviewer, which has enhanced the quality of our paper.
>
> **Requested Changes: Critical**
>
> `1) (Strengthen): Although I get how the loss arises from the PLP machinery, I don't see how PLP is useful in this paper. The background section on PLP is hard to follow and very vague. It is unclear what actual PLP language the authors are using. However, the actual loss function comes from the idea that you want at least positive items - This is very easy to express without PLP, and this is in fact done in Appendix A. Then there is no need to talk about probabilistic facts, logic rules etc.`
>
> - We understand that framing our problem as a probabilistic logic program induces additional complexity that the simplest version of our checklist may avoid. However,  we do believe that Probabilistic Logic Programming (PLP) is a powerful generalization of our approach and deserves its place in the paper. Indeed, this formalism allows us to naturally handle more complex decision rules, such as the checklist of trees. We hope that this complex, yet highly interpretable example, motivates the benefits of this approach. The flexibility brought by the probabilistic logic allows us to create any desired discrete structure, making it integral to our framework.
> Nevertheless, we understand that this additional complexity may have hurt the clarity of our manuscript. Therefore , we have revised several parts of the paper to provide a clearer motivation for PLP, better explain the terminology, and offer comparative examples of different programs, including our checklist learning program.
> - As per your suggestion, we have moved portions of the loss function from Appendix A into the main paper (Section 4.4). Additional examples have been included in Appendix A to help clarify the concepts related to PLP. We hope these changes address your concerns, and we are happy to revise further if additional clarity is needed based on feedback.
>
> `2) (Critical): In case the PLP is kept, please clarify and formalise the PLP setup`
>
> To clarify the exposition of the PLP setup, we have provided an example of probabilistic logical program (in Appendix A)  and related it to our checklist learning. We hope presenting a probabilistic logical program that is not a checklist will allow the reader to more easily grasp the general formulation.
>
> For convenience, we have copied the example below.

---

> ### Author Response · Authors · 2024-11-02
> **Author Response to Reviewer n7Qo (2/6)**
>
> (`Response 2 continued`)
>
> We consider a Probabilistic Logical Program (PLP) $\mathcal{P}$ with a simple example and then relate it to our checklist learning setup. $\mathcal{P}$ consists of a set of $N$ probabilistic facts $U = \\{U_1,...,U_N\\}$ and $M$ logical rules $F = \\{f_1,...f_M\\}$.
> - Let's first consider a non-checklist example, the well-known Alarm Bayesian problem [1]. This example features a neighborhood that can be subject to burglaries and earthquakes. Whenever such events happen, an alarm goes off in each house and people at home make a call to the emergency station.
> - In this example, the probability of a burglary occurring in the neighborhood is 0.1, and the probability of an earthquake happening in that area is 0.2. Now, let's say there are two individuals, Mary and John, who live in this locality. The probability that Mary will be home is 0.4, and the probability that John will be home is 0.5. The probabilistic facts (N=4) in this setting are:\
> U1 : P(earthquake) = 0.2 \
> U2 : P(burglary) = 0.1 \
> U3 : P(at_home(John)) = 0.5 \
> U4 : P(at_home(Mary)) = 0.4
>
> - This knowledge of the world can be easily encoded in Probabilistic Logic Programs (PLPs) using these rules (M = 3):\
> $f_1$: alarm :- burglary (a burglary will always trigger an alarm) \
> $f_2$: alarm :- earthquake. (an earthquake will always trigger an alarm) \
> $f_3$: calls(X) :- alarm, at_home(X). (if there is an alarm, and X is at home, then X calls the station)
>
> - Probabilistic programming allows to programmatically encode the knowledge of such problems and compute the probability of different queries. A possible query $q$ for this problem could be: “What is the probability that Mary makes the call?” Or, q=P(calls(Mary)).
> - While different implementations of probabilistic programming differ in their way to solve this problem, the simplest possible is to list all possible worlds that agree with the query and summing their probabilities. In this example, all possible worlds are given by the combinations of possible probabilistic facts.
> - For instance, w1 = earthquake, not burglary, not at_home(John), at_home(Mary) is a possible world. Furthermore, w1 agrees with the observation call(Mary) because there is an earthquake Mary is at home, so Mary will call the station. The probability of w1 is also P(earthquake) * (1-P(burglary)) * (1-P(John)) * P(Mary) = 0.2 * 0.9 * 0.5 * 0.4 = 0.036.
> - The final answer would sum the probabilities for all such worlds that agree with the query : \
> (earthquake, burglary, John, Mary), (not earthquake, burglary, John, Mary), (earthquake, not burglary, John, Mary), (earthquake, burglary, not John, Mary), (not earthquake, burglary, not John, Mary), (earthquake, not burglary, not John, Mary).
>
> **Moving to the checklist learning problem**
>
> - Our checklist learning setup is analogous but provides a different set of rules that combine the original probabilistic facts (concepts) into the final prediction. Suppose our dataset comprises $n$ points, and we are examining sample $i$ with its feature vector denoted as $x_i$. Our checklist has $M$ concepts in total, and the probability that each concept is true for sample $i$ is represented by the vector $p_i \in [0, 1]^M$. These constitute the $M$ probabilistic facts for sample $i$ $\\{U_1^{(i)}, …, U_M^{(i)}\\}$. The total number of probabilistic facts are $mn$. \
> For all $m \in [M]$ and $i \in [n]$, \
> $U_m^{(i)}$: P(concept $m$ is true for sample $i$) = $p_i$[m].
> - The logical rule is described by Equation 1 in the paper, which classifies a sample as positive if the number of true concepts for that sample exceeds a checklist threshold $T$. There are $n$ logical rules in total. \
> For all $i \in [n]$, \
> $f_i$: sample_positive(i) :- at_least_T_concepts_true (Sample $i$ is positive if at least $T$ concepts out of $M$ are true for sample $i$)
> - Probabilistic graphical models are constructed using these probabilistic facts and logical rules, allowing PLP to perform inference on knowledge graphs by calculating the probability of a query $q$. In our context, we query “the probability that Sample i will be classified as positive." or $q$ = P(sample_positive(i)).
> - This query is computed by summing over the probabilities of combinations that satisfy the condition $f_i$, meaning we consider all possible combinations where at least $T$ concepts are true, resulting in a positive classification.
> - Regarding our hybrid framework, the probabilistic facts are generated by a neural network (concept learner) with learnable weights, referred to as a neural predicate $U^\theta$, where $\theta$ represents the weights. These weights are trained to minimize a loss function based on the probability of a query $q$ and the ground truth. Given that we are dealing with binary classification problems, we employ Binary Cross Entropy as our loss function.

---

> ### Author Response · Authors · 2024-11-03
> **Author Response to Reviewer n7Qo (3/6)**
>
> (`Response 2 continued`)
>
> [1] Pearl, J., 2014. Probabilistic reasoning in intelligent systems: networks of plausible inference. Elsevier.
>
> We have now added the programs for learning checklists and decision trees in Appendix A.
>
> `2.a) S3: The 'probabilistic facts' are used as atoms`
>
> PLP consists of a set of ground probabilistic facts which basically encode the probability of an event (or ground atom). Each probabilistic fact corresponds to an independent Boolean random variable that is true with probability p and false with probability 1 − p.
>
> `2b) S3: The explanation of the probability of a query (the weighted model count) is very unclear. "The propagation of the realization w across the knowledge graph, according to the rules , leads to  being true". What is the knowledge graph? How are the rules defined? Are those horn clauses?`
>
> The above example explains how rules can be defined for each problem.
> All possible worlds are given by the combinations of possible probabilistic facts. These worlds can be represented as different paths in a graph. We traverse the graph to compute the probability of a specific query (defined by a subset of these worlds/paths).
>
> `2c) S4.4: The goal of PLP is to create a program, but I see none in the method. What is the actual program used to encode the checklist? Eq 2 talks about some  that supposedly encodes the logic program but never defines it. And for the decision tree learning?`
>
> We have explained the program for Checklist learning above (Response 2). This has also been included in Appendix A.2. Additionally, we have provided the program for Decision Trees in Appendix A.3.
>
> `3) (Critical): Appendix A should be part of the main text. It's a simple and short argument that was critical in helping me understanding what was going on.`
>
> We thank the reviewer for this suggestion. We have now moved parts of Appendix A to Section 4.4 in the main paper.
>
> `4) (Critical): Especially if PLP is kept, the comparison to DeepProbLog (Manhaeve et al. 2018) should be more thorough, as ProbCheckList can be directly encoded in DeepProbLog.`
>
> We appreciate the reviewer's suggestion. We have now explicitly stated that our focus is on specific instantiations of DeepProbLog, driven by real-world applications. Additionally, we have clarified that DeepProbLog offers an innovative approach for learning interpretable, discrete structures, such as checklists and decision trees. We have now included this in Related Works (Section 2).
>
> For convenience we have copied it here:
>
> *Methods like DeepProbLog offer an innovative approach for learning interpretable, discrete structures, such as checklists and decision trees. We study specific instantiations of these methods that are driven by real-world applications.*
>
> `5) (Critical): The complexity analysis can be significantly improved. There are papers that use dynamic programming to compute this in polytime [1, 2, 3] for the case where the number of true concepts is exactly, these need to be discussed. It should be possible to scale far beyond M=30 with these by doing this DP algorithm for everything above`
>
> - We appreciate the reviewer for directing us to these papers. Incorporating techniques like k-subset sampling could indeed help scale the method beyond M=30.
> - Furthermore, it's important to highlight that a significant contribution of this work is its time-efficient implementation using tensorization. While the naive exponential-time implementation was impractically slow, taking days to train, our approach drastically reduces training time to minutes by utilizing more memory.
> - Drawing inspiration from papers [1] and [3], relaxed sampling methods can be designed to approximate the top concepts and reduce memory requirements. We've now included this discussion in the Limitations section (Appendix L), as it offers a promising direction for real-world applications of ProbChecklist. We sincerely thank the reviewer for bringing this relevant work to our attention.
>
> `6) (Critical): Concepts that are trained end-to-end without direct supervision like in this paper will have reasoning shortcuts [4]. A short discussion on this limitation is important.`
> - We thank the reviewer for sharing this reference, and we've now incorporated it into our discussion on the interpretability of learned concepts in Section 6.
> - While visual inspection can indeed be challenging due to its laborious nature, need for domain experts and may lead to inaccurate pattern recognition. We would like to emphasize an important point: in our MNIST checklist learning setup, we defined rules for the entire image, i.e. Image 1 should be even, but fixed the learning to four concepts per image. While deciphering each concept individually proved difficult, the combination of all four concepts accurately represented the image-level concept. This indicates that the model successfully captured granular details. It is hard to predefine individual concepts for each image beforehand.

---

> ### Author Response · Authors · 2024-11-03
> **Author Response to Reviewer n7Qo (4/6)**
>
> (`Response 6 continued`)
>
>
> For convenience we have copied the excerpt here:
>
> *It has been shown that Neuro-Symbolic models trained end-to-end often fall prey to reasoning shortcuts, leading them to learn concepts with unintended meanings to achieve high accuracy. This makes it harder to interpret the learnt concepts. We encourage practitioners to adopt existing methods [4] that help mitigate these issues.*
>
> **Requested Changes  - Smaller Points**
>
> `7) In Figure 1, what do \oly, \ymph etc represent?`
> - In Figure 1 \oly, \ymph, etc represent tokens. For the Neoplasm detection task using clinical notes (text), we conducted a token frequency analysis. We identified key tokens associated with positive and negative concepts (positive and negative tokens). Each concept is defined by the presence of positive words and the absence of negative words. Further details about this can be found in Section 5.
> - We have now mentioned this in the caption of Figure 1 to improve clarity.
>
> `8) Sec 2: "Our method does not rely on integer programming and thus exhibits much faster computing times". I think this is quite arguable, considering that the method requires computing the loss function that, until now, cannot be scaled beyond M=30.`
>
> We provide support for this observation through experimental results presented in Table 2 of Appendix B, where we compare the training times (in seconds) for both MIP Checklist and ProbChecklist. As noted in the paper, "*While MIP Checklist performs well on tabular datasets, consistently finding optimal solutions, it struggles significantly with the MNIST synthetic setup. Even when we allow the Gurobi solver to run for 1 hour, it fails to find optimal solutions in most cases. In contrast, ProbChecklist proves to be more reliable, handling end-to-end training effectively and consistently learning the optimal solution.*"
>
> `9) Sec 3: I don't understand the binary mapping $\phi_M 1_k =1 1_k$ . Shouldn't the output be $1_M$?`
>
> Yes, the binary mapping is defined such that the sum of each row of $\phi_m$ should be 1, i.e., $\phi_M 1_k =1 1_M$. We thank the reviewer for pointing this out and we have now corrected it in the paper.
>
> `10) Sec 4: I got confused by the hyperparameter d_k’. I think it could have more emphasis in how it's defined.`
>
> For each modality, we use distinct concept extractors (such as neural networks) to learn an embedding, which outputs soft concept probabilities. The dimension of the embedding learned from the k-th modality is denoted as $d_k'$. We define $d’ = \sum_{k=1}^K d_k’$. We have ensured that this is defined clearly in Section 4.3.
>
> `11) Sec 4: The distinction between q, y_i and \Hat{y_i} is lost on me.`
> - In Probabilistic Logic Programming (PLP), $q$ denotes the query. For our checklist learning setup, we use the query, "Is the sample classified as positive?" This corresponds to calculating the probability that at least $T$ out of $M$ concepts are positive. For clarification on PLP terminology, please refer to Response 2 and Appendix A.
> - $y_i$ represents the ground truth label for sample $i$ (indicating whether the sample is positive or not), and $\hat{y_i}$ denotes the predicted label. $y_i$ is defined at the beginning of Section 3 and $\hat{y_i}$ is defined in Equation 1.
>
> `12) Sec 4.3.1: The 'focused concept extractor' just seems like feature selection to me?`
>
> Yes, focused concept learners can indeed be viewed as a feature selection approach that enhances interpretability. We propose using distinct concept learners for each modality to make the learned concepts more interpretable, rather than combining multiple modalities into complex, less human-understandable features.
>
> `13) Sec 4.4: Eq 2 is a lot easier to understand as it's posed in the appendix. Currently without a clear definition of , it's very hard to follow.`
>
> We have now explained the PLP terminology better with respect to the checklist learning example and incorporated parts from Appendix A into the main paper. Additionally, we have included more examples of PLP in Appendix A. We thank the reviewer for this suggestion.

---

> ### Author Response · Authors · 2024-11-03
> **Author Response to Reviewer n7Qo (5/6)**
>
> `14) Sec 4.4: Proposition 4.1 is not formal enough. There is no clear statement of the assumptions or the program from which this result follows. Furthermore, the $\sigma$  and $\sum_d$ seem somewhat like superfluous notation for the purposes of this paper and is not used anywhere else.`
>
> We have now added the checklist learning program in Appendix A from which Proposition 4.1 follows. Based on the reviewer's suggestion we have moved the loss function from equation from Appendix A to the where the notation $\sigma$  and $\sum_d$ is used. We hope this will address your concern.
>
> `15) Sec 4.7: Eq 6 seems like the last term should be not [2,2]?`
>
> We thank the reviewer for pointing this out. We have corrected it now.
>
> `16) Sec 4.7: It's not clear to me why balanced trees would 'provide a faithful representation of data, fostering diversity and interpretability of the learned concepts'`
>
> When we initially trained simple decision trees without enforcing the balanced tree constraint, we observed that some of the learned concepts were trivial (either entirely true or false). These spurious features led to imbalanced trees, compromising their interpretability and utility. To address this, we introduce entropy-based regularization techniques aimed at learning balanced decision trees. The intuition behind them is that each node will learn features that effectively split the samples into two significantly sized groups. This promotes interpretability by ensuring that each node's decision is meaningful rather than trivial.
>
> Without this regularization, the tree could rely on very few concepts (or maybe even a single concept) that perfectly predict the true class, while assigning all remaining nodes are always true or false. Such a tree lacks interpretability. By enforcing balance, we ensure that most nodes contribute to learning meaningful and interpretable concepts. This explanation has now been added to Appendix K.
>
> `17) Sec 5: I don't understand how MIP and ILP are used. Those aren't learning algorithms, right?`
>
> - ILP and MIP are existing checklist learning methods which rely on integer programming (not learning algorithms). These form as key baselines for our method.
> - We refer the reviewer to Section E.1 where additional details about these methods are given.
>
>
> `18) Sec 5: I think it should be highlighted that the MNIST problem can be solved with $d_k’ =1$, suggesting that the algorithm isn't learning the right / interpretable concepts since $d_k’ = 4$  is needed.`
>
> We have shown the performance for various values of $d_k'$ in Figure 4a. The results are discussed in the 'Sensitivity Analysis' point (Section 5.1) and further elaborated in Appendix E.6. As stated in the paper, "We observe a marked performance improvement when $d'_k$ increases from 1 to 2, with saturation occurring after $d'_k = 3$. This indicates that learning only one concept per image is insufficient to capture the full range of signals in the sample." While a higher embedding dimension better captures the relevant patterns for the classification task, it does not necessarily imply that the learned features for higher $d_k'$ will be more interpretable. Depending on the task and dataset, we would need to find suitable interpretability techniques to check how interpretable the concepts are.
>
> `19) Sec 5.1: Table 1: I don't understand why there are error bars under T. I thought this was a hyperparameter?`
>
> - It is important to note that T is not a hyperparameter, but rather a parameter used to define the structure of the checklist. Given a fixed list of M binary concepts, samples are classified based only on the parameter T (Equation 1). Users need to optimize over the parameter T to find the most performant checklist classifier. We report the standard deviation of the best T observed across 5 runs. We would be happy to answer further questions about this.
> - Appendix C contains a list of all the hyperparameters and given strategies to find optimal values.
>
> `20) Sec 5.1: Table 1: Under precision - PhysioNet Tabular, ProbCheckList is bolded but LR has a higher score.`
>
> This is an inadvertent oversight on our part and we have fixed this error in the new revision. We thank the reviewer for pointing it out.

---

> > ### Author Response · Authors · 2024-11-03
> > **Author Response to Reviewer n7Qo (6/6)**
> >
> > `21) Sec 5.1: The order in which the datasets are discussed is different from Table 1`
> > - We have changed the order of the datasets to match with Table 1.
> > - We thank the reviewer for noticing such fine details that improve the paper.
> >
> > `22) Sec 5.2: I understand why this is studied, but I don't think this feature attribution method relying on 'visual inspection' is a thorough and infallible way to introduce interpretability...`
> > - Our final classifier, a discrete checklist, is highly interpretable. Although ProbChecklist does require interpretable concept extractors, we are fortunate to be able to rely on existing work. While this reliance might be viewed as a limitation, it’s important to highlight that this is the first method capable of learning checklists from complex modalities.
> > - We encourage practitioners to explore alternative interpretability frameworks better suited to their specific domains. For MNIST, as discussed in Section 5.2, the analysis depends primarily on visual inspection. In addition, we offer a more robust approach for text data by analyzing token frequency, which results in a human-readable checklist of positive and negative tokens (Figure 1).
> >
> >
> >
> > **New Revision**
> >
> > We have incorporated your feedback in the new revision (all changes are highlighted in red). The changes are:
> > - Updated the caption of Figure 1.
> > - Added a comparison of our technique with DeepProbLog in Related Works.
> > - Corrected the typing mistake in the notation for binary mapping in Section 3.
> > - Elaborated on the hyperparameter d_k’ in Section 4.3
> > - Moved parts of the proof of Proposition 4.1 from Appendix A to Section 4.4
> > - Corrected the typing mistake in Equation 6 (Decision Tree Logic Rule)
> > - Reordered the datasets in Section 5.1 to match with the order in Table 1
> > - Updated the discussion on interpretability of concepts in Section 6 to talk about reasoning shortcuts in Neuro-Symbolic models.
> > - Updated Appendix A to include an example of Probabilistic Logic Programming, the program used for learning checklists and decision trees.
> > - Added a discussion on why balanced trees enhance interpretability of the concepts in Appendix K.2
> > - Included a discussion on k-subset sampling in Appendix L (Limitations).
> >
> > ---
> > Thank you again for reviewing our paper and for your encouraging feedback!

---

### Review · Reviewer_asA3 · 2024-12-05

**Summary Of Contributions:**

Authors propose a novel method called "ProbChecklist" for creating interpretable predictive checklists from diverse modalities. They introduce a method to learn checklists from various data types, addressing limitations of previous works restricted to given data types. This leverages probabilistic logic programming for learning binary concepts from high-dimensional inputs. Authors perform extensive experiments showing superiors performance compared to state-of-the-art methods. These experiments suggest that the methodology can be adapted to healthcare domain, especially in the context of textual reports, improving interpretability and fairness.

**Audience:**

Yes

**Claims And Evidence:**

Yes

**Requested Changes:**

- Provide examples of how domain experts (e.g., clinicians) can validate learned concepts to ensure consistency.
- Investigate distributed or parallelized training frameworks to reduce computational bottlenecks.
- Ablation study to compare checklist variants (e.g., weighted checklists or those with decision trees) to guide readers on their use in different scenarios.

**Strengths And Weaknesses:**

Strengths:

- ProbChecklist enables the creation of predictive checklists from multiple modalities, which improves its applicability beyond the usual tabular data.
- Fairness constraints allows the framework to balance the trade-off between performance and fairness.
- Probabilistic logic programming integrated with gradient-based deep learning methods offere faster computation and better optimization compared to classic integer programming-based approaches.

Weaknesses:

- While the checklist structure itself is interpretable, high-dimensional inputs may be difficult to interpret, requiring additional efforts to explain their meaning.
- While the method avoids integer programming, the use of probabilistic logic programming can become computationally expensive for very large datasets or highly complex input modalities, potentially hindering scalability.
- Although the fairness constraints reduce disparities, they do not guarantee strict fairness across all subgroups, and their effectiveness heavily depends on hyperparameter tuning.
- The proposed "checklist of decision trees" adds interpretability at the cost of additional computational complexity, making the model harder to train effectively compared to simpler architectures.

---

> ### Author Response · Authors · 2024-12-12
> **Author Response to Reviewer asA3 (1/2)**
>
> We sincerely thank the reviewer for their constructive feedback, which has greatly contributed to improving our paper.
>
> **Strengths**
>
> We particularly thank the reviewer for recognizing the strengths of our approach, in terms of its broader applicability, its fairness regularization, and its computational advantages. Below, we provide in-depth answers to the questions raised by the reviewer.
>
> **Weaknesses**
>
> `1. While the checklist structure itself is interpretable, high-dimensional inputs may be difficult to interpret, requiring additional efforts to explain their meaning.`
>
> Although interpretable concept extractors are required for our method, we are fortunate to rely on the rich existing literature in this area. We want to emphasize that our primary objective and motivation was to study the feasibility of checklist learning from complex data modalities. We therefore relied on existing work for providing interpretable concept learners. We have now detailed this trade off in the discussion on the interpretability of the learned concepts (Section 6)
>
>
> `2. While the method avoids integer programming, the use of probabilistic logic programming can become computationally expensive for very large datasets or highly complex input modalities, potentially hindering scalability.`
>
> - We agree with the reviewer that probabilistic logic programming (PLP) can be computationally expensive for very large datasets, and further exploration of this issue is a promising direction for future work. However, it's important to note that PLP represents a significant improvement over integer programming. Our experimental results, shown in Table 2 of Appendix B, compare training times (in seconds) for both MIP Checklist and ProbChecklist. As noted in the paper, "While MIP Checklist performs well on tabular datasets and consistently finds optimal solutions, it struggles significantly with the MNIST synthetic setup. Even when we allowed the Gurobi solver to run for an hour, it failed to find optimal solutions in most cases. In contrast, ProbChecklist handled end-to-end training more reliably and consistently learned the optimal solution."
>
> - Moreover, a key contribution of our work is its time-efficient implementation using tensorization. The naive exponential-time implementation was impractically slow - taking days to train, but our approach significantly reduces training time to minutes by leveraging more memory.
>
>
> `3. Although the fairness constraints reduce disparities, they do not guarantee strict fairness across all subgroups, and their effectiveness heavily depends on hyperparameter tuning.`
>
> We agree with the reviewer’s point. The fairness regularizer does require adjusting the relative weights for different subgroups to achieve desired fairness levels. This can actually be beneficial, as some applications may not require strict fairness across all subgroups but instead prioritize fairness for a specific group. This flexibility allows the model to focus on fairness where it is most needed.
>
> `4. The proposed "checklist of decision trees" adds interpretability at the cost of additional computational complexity, making the model harder to train effectively compared to simpler architectures.`
>
> We acknowledge the trade-off between computational complexity and interpretability identified by the reviewer. Yet, one objective of the checklist of decision trees is to illustrate the versatility of the PLP formalism.

---

> ### Author Response · Authors · 2024-12-12
> **Author Response to Reviewer asA3 (2/2)**
>
> **Requested Changes:**
>
> `1. Provide examples of how domain experts (e.g., clinicians) can validate learned concepts to ensure consistency.`
>
> - For continuous data, identifying concepts is straightforward as they are defined by the thresholds used for binarization. For instance, Figure 3 shows examples like the mean, standard deviation, and last value of medical time series data, which clinicians can verify based on domain knowledge to validate the final checklist.
>
> - In the NLP setting, we define concepts by the presence or absence of specific tokens in the text. Figure 1 contains key tokens (for neoplasm detection), derived from patient symptoms or medical reports, which can indicate the final diagnosis.
>
> - In the time series setting, we use gradient attribution analysis to highlight regions of the series that are positively or negatively activated for a concept in a patient. By repeating this for multiple patients with positive and negative outcomes, clinicians can identify trends like fluctuations, monotonicity, or constancy in the final hours, which assist in constructing checklists. A small example is provided in Appendix F.4 (Figure 10). This method relies heavily on visual inspection, but since ProbChecklist is flexible, more interpretable concept extractors can be easily integrated using existing methods.
>
> - We have included this discussion in Appendix F.6
>
> `2. Investigate distributed or parallelized training frameworks to reduce computational bottlenecks.`
>
> We sincerely thank the reviewer for this suggestion. We've now included this discussion in the Limitations section (Appendix L), as it offers a promising direction for real-world applications of ProbChecklist. An active area of research focuses on parallelizing the loss (https://arxiv.org/abs/2112.00364) . Drawing inspiration from similar papers, we can reduce computational requirements.
>
> `3. Ablation study to compare checklist variants (e.g., weighted checklists or those with decision trees) to guide readers on their use in different scenarios.`
>
> The main goal of introducing checklists of decision trees is to show the flexibility of the probabilistic logic programming (PLP) framework. While both checklists and decision trees require users to specify the number of concepts based on domain knowledge, decision trees add complexity due to the need to search over tree structures, introducing extra hyperparameters like tree depth and node placements. Given the computational intensity, we limited decision tree experiments to synthetic datasets, where we designed a true checklist of decision trees to validate whether the recovered structure matches the ground truth.
>
> **New Revision**
>
> In this revision, we have incorporated the feedback provided by Reviewer asA3, and all changes are highlighted in green. The changes are:
> - Expanded the limitations section (Appendix L) of the paper to include parallel training as a future direction.
> - Added Appendix F.6 to discuss how domain experts can validate the learnt concepts.
>
> ---
>
> Thank you again for reviewing our paper and for your encouraging feedback!

---

### Decision · Action_Editor_VpCc · 2025-01-15

**Recommendation:** Reject

**Comment:**

All three reviewers are in agreement that the work, if revised, could potentially present a set of modified claims and corresponding justifications. This would involve expanding the experimentation towards real data that the authors currently consider out of scope for their manuscript as is. As such, I encourage the authors to consider integrating these reviewer comments in considering an expanded iteration of their manuscript.

**Audience:**

As a TMLR submission, I think it would benefit from a more thorough justification of PLP and a an expanding set of arguments, theory and/or experiments to justify the tunable interpretability claims of the proposed method. There may be another academic community where this methodology is more established and accepted, but for TMLR's audience, the manuscript would benefit from a major revision to align and justify the foundation for the work.

**Claims And Evidence:**

All three reviewers have evaluated the manuscript and its revision. They have discussed the authors' feedback and each other's viewpoints. The majority of reviewers continue to voice concerns around depth of evaluation and are not satisfied by the evidence to support the manuscripts claims. In particular, the interpretability of the proposed method, along with assumptions around the established acceptance of PLP remain common themes of concern. The revision was primarily focused on additions to the appendix, which the reviewers found insufficient in addressing their concerns around justifying the claims made in the main text of the paper.

**Resubmission Of Major Revision:**

The authors may consider submitting a major revision at a later time.